# Partially sintered copper–ceria as excellent catalyst for the high-temperature reverse water gas shift reaction

Hao-Xin Liu [1,4], Shan-Qing Li [2,4], Wei-Wei Wang [1], Wen-Zhu Yu [1], Wu-Jun Zhang [3], Chao Ma [3✉] & Chun-Jiang Jia [1✉]

For high-temperature catalytic reaction, it is of significant importance and challenge to construct stable active sites in catalysts. Herein, we report the construction of sufficient and stable copper clusters in the copper–ceria catalyst with high Cu loading (15 wt.%) for the high-temperature reverse water gas shift (RWGS) reaction. Under very harsh working conditions, the ceria nanorods suffered a partial sintering, on which the 2D and 3D copper clusters were formed. This partially sintered catalyst exhibits unmatched activity and excellent durability at high temperature. The interaction between the copper and ceria ensures the copper clusters stably anchored on the surface of ceria. Abundant in situ generated and consumed surface oxygen vacancies form synergistic effect with adjacent copper clusters to promote the reaction process. This work investigates the structure-function relation of the catalyst with sintered and inhomogeneous structure and explores the potential application of the sintered catalyst in C1 chemistry.

[1] Key Laboratory for Colloid and Interface Chemistry, Key Laboratory of Special Aggregated Materials, School of Chemistry and Chemical Engineering, Shandong University, Jinan 250100, China. [2] Key Laboratory of Micro-Nano Powder and Advanced Energy Materials of Anhui Higher Education Institutes, Chizhou University, Chizhou 247000, China. [3] College of Materials Science and Engineering, Hunan University, Changsha 410082, China. [4] These authors contributed equally: Hao-Xin Liu, Shan-Qing Li. ✉email: cma@hnu.edu.cn; jiacj@sdu.edu.cn

Supported metal catalysts have been widely used in the industrial catalysis process because of their adequate active sites and high atom utilization[1–5]. Recently, nanoengineering has been widely applied on the preparation of solid catalysts with a homogeneous surface structure by anchoring active metal on stable supports[6,7]. Maintaining the uniformity and high dispersion of the active metal sites has been considered to be the key to the excellent activity of catalysts[8]. However, with the sintering of catalysts, the active metal tends to agglomerate especially under the reaction conditions of high temperature and reducing atmosphere, leading to the serious deactivation[9,10]. Reducing the loading of the active metal can partly overcome the aggregation, however, the activity of the catalyst is commonly unsatisfactory due to the insufficient active metal sites[11]. Thus, the construction of stabilized and adequate active sites in the sintered catalyst is a great challenge but also full of significance.

The reverse water gas shift (RWGS) reaction is recognized a most promising way to utilize $CO_2$, thanks to its high selectivity and low operation pressure[12–15]. The resulting CO is considered as feedstock to produce various value-add chemicals via Fischer-Tropsch synthesis or other syngas process[16–18]. Due to its endothermic property, a high working temperature is usually required to facilitate the equilibrium conversion of $CO_2$. However, it brings huge difficulty to the durability of the catalysts under such harsh conditions[14,19,20]. Noble metal catalysts, such as Pd- and Pt- based catalysts[11,21,22], have been studied for this reaction, however, their practical applications are limited by the inferior catalytic performance and high cost. Among the non-noble metal catalysts, Cu-based catalyst has been considered as the ideal candidate for this reaction because of the high activity, selectivity, and low cost[13,14,23,24]. However, on one hand, the catalysts with high copper loading are apt to agglomeration, causing severe deactivation[24]. On the other hand, the low copper loading on supports can partly resist aggregation, but suffers from insufficient active sites and poor catalytic performance[7]. As a result, the conflict between high activity and high stability under harsh reaction conditions limits the development and application of Cu-based catalysts. According to the previous work, ceria $(CeO_2)$ is often recognized as a suitable supports to anchor copper because of the strong interaction between copper and ceria. Using the interaction, the copper–ceria catalyst has shown its unique value in a variety of catalytic reactions, such as low-temperature water-gas shift reaction[25,26], CO oxidation[27], and $CO_2$ hydrogenation[23,28]. Besides, $CeO_2$ tends to sintered after high-temperature aging, resulting in a dramatic structural transformation[29]. Recently, our group has constructed stable and atomically dispersed copper site with unsaturated coordination in the sintered copper–ceria catalyst with very low Cu loading of 1 wt% by air-calcination at 800 °C, which exhibited very high and stable activity for CO oxidation[27]. However, during the long-term harsh reaction conditions of high temperature and reductive atmosphere, the structure of the copper–ceria catalyst is still unclear, especially for the catalyst with relative high copper loading, which undoubtedly limits the development and application of the copper–ceria catalyst.

Herein, we report a partially sintered Cu/$CeO_2$ catalyst with Cu loading up to 15 wt.% which exhibits extraordinarily high activity and stability for the RWGS reaction under very harsh conditions (600 °C, space velocity of 400,000 mL·$g_{cat}^{-1}$·$h^{-1}$). Two-dimensional (2D) and three-dimensional (3D) copper clusters are formed and firmly anchored on the surface of ceria under the reaction conditions due to the interaction between copper and ceria, through which abundant stable active sites were constructed. Further, structural characterization and DFT calculations confirmed that abundant active surface oxygen vacancies were in situ generated and consumed circularly during the reaction, which combined with adjacent copper clusters to promote the activation of $CO_2$ and catalytic efficiency. The synergistic catalytic effect of anti-sintering active copper clusters and sufficient surface oxygen vacancies provided a guarantee for the extraordinary activity and stability under harsh conditions. The partially sintered catalyst with excellent catalytic performance breaks the conventional impression that catalysts are severely deactivated upon sintering and shows great potential in the utilization of $CO_2$.

## Results

**Catalytic performance in the RWGS reaction.** The catalytic performance in the RWGS reaction over various catalysts was evaluated at various temperatures under a high space velocity of 400,000 mL·$g_{cat}^{-1}$·$h^{-1}$. As shown in Fig. 1a, the $CeO_2$ support itself showed very poor catalytic activity. When the reaction temperature reached 600 °C, $CO_2$ conversion was only 8%. While, after depositing copper on ceria, the catalyst significantly promoted the catalytic activity. The catalytic activity increased with the increasing of copper loading until the copper loading reached 15 wt.% (Supplementary Fig. 1). Among all the prepared catalysts, the 15CuCe catalyst showed the best activity of 146.6 $mol_{CO2}$·$g_{cat}^{-1}$·$s^{-1}$ at 600 °C, which was at least three times higher than all the other reported catalysts. In addition, the reaction rate of 15CuCe was as high as 52.2 $mol_{CO2}$·$g_{cat}^{-1}$·$s^{-1}$ at 500 °C, more than one order of magnitude than that of other reported non-noble metal catalysts and even noble metal catalysts (Fig. 1b and Table 1). And it is worth noting that these Cu catalysts show 100% selectivity of CO with no $CH_4$ detected in the products (Supplementary Fig. 1b). Besides, the catalytic performances of the 15CuCe catalyst under other reaction atmospheres with different $H_2$:$CO_2$ ratios were also evaluated. As shown in Supplementary Fig. 2, the $CO_2$ conversion increased with the increasing of the $H_2$:$CO_2$ ratio. And at all these $H_2$:$CO_2$ ratios, the 15CuCe catalyst shown excellent catalytic activity, suggesting this catalyst had high catalytic efficiency over a wide range of $H_2$:$CO_2$ ratios. And it was noteworthy that even when the $H_2$:$CO_2$ ratio reached 4:1, no $CH_4$ was detected in the production, which indicated the catalyst catalyzed the RWGS reaction much rather than the methanation. To the best of our knowledge, the excellent activity of the 15CuCe catalyst at high temperature is unmatched. As shown in Fig. 1c and Supplementary Fig. 3, the apparent activation energy $E_a$ of the 5CuCe catalyst (62.88 kJ·$mol^{-1}$) and the 15CuCe catalyst (57.92 kJ·$mol^{-1}$) was much lower than that of the 15CuAl catalyst (133.96 kJ·$mol^{-1}$). Besides, the 15CuCe catalyst synthesized with $CeO_2$ nanorod as support exhibited much better activity than that of the 15CuCe-NC and 15CuCe-NP catalysts, indicating the morphologies of $CeO_2$ supports had important effect on the catalytic performance (Supplementary Fig. 4).

Long-term evaluation of the copper–ceria catalysts was conducted. As illustrated in Fig. 1d, both the 5CuCe and 15CuCe catalyst showed excellent stability. For the 5CuCe catalyst and the 15CuCe catalyst, even under the very harsh condition (600 °C, GHSV = 400,000 mL·$g_{cat}^{-1}$·$h^{-1}$), it maintained more than 85% of its initial activity after 70 h test. The activity of both the 5CuCe and the 15CuCe catalysts decreased gradually in the first 20 h and then stabilized. In order to explore the stability of the 15CuCe catalyst further, the test time was extended to 240 h, and the 15CuCe catalyst still showed excellent stability. However, the 15CuCe-NP catalyst with more severe sintering (Supplementary Table 1) showed worse stability and lower activity in the 40 h test (Supplementary Fig. 5). And the reference 15CuAl catalyst lost more than 50% of its original activity within 10 h under the same test conditions. In addition, the reaction rate of the 15CuCe catalyst decreased slightly after six rounds of start-up cool-down stability tests (Supplementary Fig. 6). From the above results, the

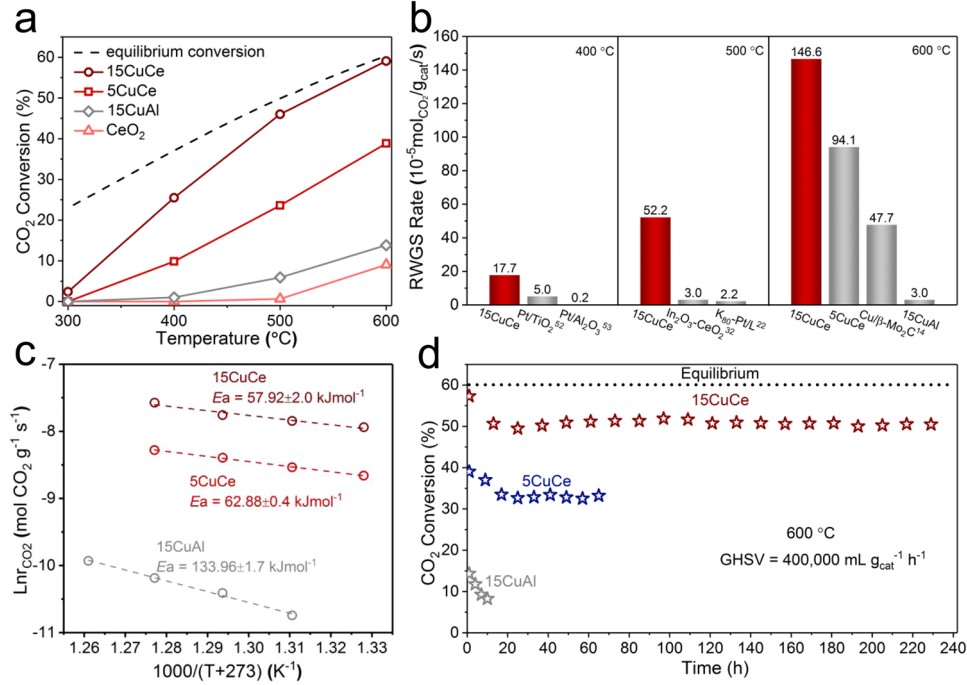

**Fig. 1 Catalytic performance of copper–ceira catalysts in RWGS reaction. a** Activities of $CeO_2$, 5CuCe, 15CuCe, and 15CuAl catalysts; **b** Comparison of reaction rates for different catalysts at 400 °C, 500 °C, and 600 °C; **c** Apparent activation energy value of various catalysts; **d** Long-term catalytic tests of the 5CuCe, 15CuCe catalysts and the reference 15CuAl catalyst.

**Table 1 Comparison of $CO_2$ Conversion Rate and CO Selectivity for the as-prepared and Literature Reported catalysts.**

| Catalyst | $H_2$:$CO_2$ | Temperature (°C) | Pressure (MPa) | Rate ($10^{-5} mol_{CO2}/g_{cat}$/s) | CO selectivity (%) | Ref |
|---|---|---|---|---|---|---|
| 15CuCe | 3:1 | 600 | 0.1 | 146.6 | 100 | this work |
| 15CuAl | 3:1 | 600 | 0.1 | 3.0 | 100 | this work |
| $Cu/CeO_2$–hs | 3:1 | 600 | 0.1 | 42.5 | 100 | 46 |
| $4Cu–Al_2O_3$ | 2:1 | 600 | 0.1 | 17.9 | 100 | 13 |
| $Cu/\beta–Mo_2C$ | 2:1 | 600 | 0.1 | 47.7 | 99.2 | 14 |
| $Cu-Fe/SiO_2$ | 1:1 | 600 | 0.1 | 11.9 | 100 | 20 |
| NiCe/Zr | 3:1 | 550 | 0.1 | 33.3 | 99.5 | 48 |
| 15CuCe | 3:1 | 500 | 0.1 | 52.2 | 100 | this work |
| $In_2O_3–CeO_2$ | 1:1 | 500 | 0.1 | 2.98 | 100 | 49 |
| $K_{80}–Pt/L$ | 1:1 | 500 | 0.1 | 2.22 | 100 | 22 |
| Ni–in–Cu | 3:1 | 500 | 0.1 | 3.95 | 100 | 50 |
| $CuSiO/CuO_x$ | 3:1 | 500 | 0.1 | 3.18 | 100 | 51 |
| $TiO_2/Cu$ | 3:1 | 500 | 0.1 | 1.78 | N/A | 51 |
| $SiO_2/Cu$ | 3:1 | 500 | 0.1 | 1.11 | N/A | 51 |
| Cu–Zn–Al | 2:1 | 500 | 0.1 | 26.1 | 100 | 14 |
| $Cu/\beta–Mo_2C$ | 2:1 | 500 | 0.1 | 37.9 | 99.0 | 14 |
| $Pt/TiO_2$ | 1:1 | 400 | 0.1 | 5.0 | 100 | 52 |
| $Pt/Al_2O_3$ | 1.4:1 | 400 | N/A | 0.16 | N/A | 53 |
| Ni/Mg(Al)O | 3:1 | 450 | 0.1 | 0.5 | 66.7 | 54 |
| $Fe-CeO_2$ | 4:1 | 400 | 0.1 | 0.65 | 100 | 55 |

optimal 15CuCe catalyst achieved the combination of high activity and high stability.

**Structural characterization of the copper–ceria catalysts.** The catalytic performances of catalysts are closely related to the structure of the catalyst. The transmission electron microscopy (TEM) images of the fresh 15CuCe catalyst presented the regular rod-like structure with length ranging from about 50 to 200 nm (Supplementary Fig. 7a). The EDS mapping results (Supplementary Fig. 7b) suggested that copper species were well dispersed in the fresh 15CuCe catalyst. The catalysts with relative

low copper loading exhibited the similar size and morphology as the 15CuCe catalyst (Supplementary Fig. 8). As for the fresh and used 25CuCe catalysts, a large number of isolated copper particles (tens to hundreds nanometers, labeled by circle in red) were observed (Supplementary Fig. 8h and k), which indicated that the optimized copper loading was around 15 wt.%. This phenomenon showed that it was crucial to load proper amount of copper on the ceria support. The addition of excess copper results in agglomeration and deactivation, while catalysts with low Cu concentration suffer from insufficient Cu active sites, ending up with poor catalytic performance. For the 15CuCe catalyst after $H_2$ pretreatment, the high-angle annular dark-field (HAADF) images

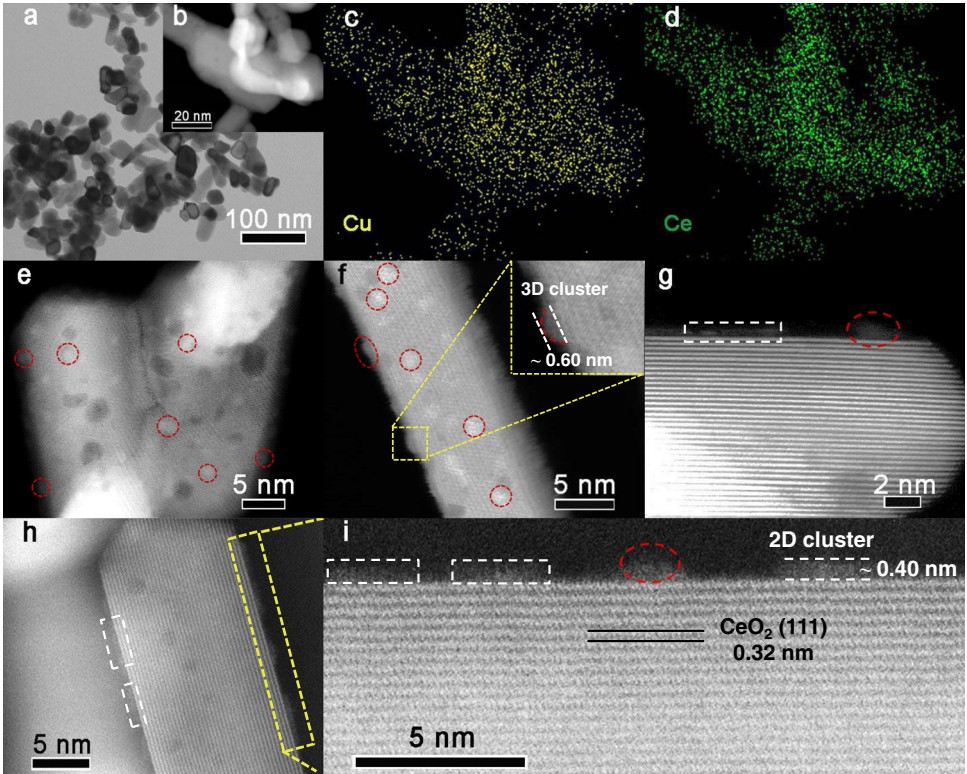

**Fig. 2 Structure characterization of the 15CuCe catalyst after 70 h stability test. a** Transmission electron microscope (TEM) image; **b–d** Scanning transmission electron microscope (STEM) image and element mapping results; **e–h** High-angle annular dark-field (HAADF) STEM images; **i** Enlarged image of the highlighted region in (**h**).

(Supplementary Fig. 9) indicated that ceria nanorods could maintain the rod-like morphology, suggesting ceria nanorods have not undergone obvious sintering prior to the RWGS reaction. In addition, 2D layered and 3D hemisphere-shaped copper clusters could be clearly observed on the surface of $CeO_2$ nanorods. The width of the 2D layered clusters ranged from 1 nm to 3.5 nm, and the thickness was from about 0.2 nm to 0.5 nm. The diameter of 3D clusters was around 2 nm. Besides, as shown in Fig. 2a, compared to the used catalyst with slight sintering after temperature-dependent evaluation (Supplementary Fig. 8j), the $CeO_2$ nanorods underwent more obvious sintering during the long-term reaction of 70 h at high temperature. However, the excellent activity and stability of the catalyst (Fig. 1a and d) meant that there might still be abundant active metal sites on the partially sintered ceria support. As shown in Fig. 2b–d and Supplementary Fig. 10, the EDS elemental mapping images demonstrated the high dispersion of copper, with Cu signal appearing uniformly on the surface of the partially sintered catalyst. And the HAADF images indicated on the surface of the partially sintered copper-ceria catalyst, copper also existed dominantly in the forms of 2D layered clusters and 3D hemisphere shaped clusters (Fig. 2e–i and Supplementary Fig. 11), similar to the activated samples. The average thickness of the layered clusters was about 0.4 nm (Fig. 2i), which was approximately consistent with a bilayer configuration of copper atoms[25]. The widths of the 2D layered clusters varied from 1.5 nm to 4 nm. And the mean width and average thickness of the 3D clusters were 1.3 nm and 0.6 nm, respectively.

Although many excellent reports have explored the structure of copper-ceria catalysts[25,27], the structure of the copper-ceria catalyst, including the status of copper species and morphology of ceria, is almost unknown under harsh reaction conditions (at high temperature and with the reductive atmosphere), especially

for the catalyst with relatively high copper loading. In order to explore the structure of the 15CuCe catalyst after a long enough reaction time, the measurement by an HAADF-STEM of the sample after 240 h RWGS reaction was performed. As shown in Fig. 3a, $CeO_2$ nanorods were still partially sintered, but not completely sintered, which indicated that $CeO_2$ nanorods could not oversinter to cause the severe deactivation. And the EDS mapping images (Fig. 3b–d and Supplementary Fig. 12) suggested that the copper species were still highly dispersed on the partially sintered $CeO_2$ support. As illustrated in Fig. 3e–i, even though the reaction time has been extended to 240 h, 2D and 3D copper clusters were anchored on the partially sintered $CeO_2$ nanorods, which undoubtedly prevented the catalyst from being inactivated by the agglomeration of active copper species. According to the above experimental results, the structural evaluation of the 15CuCe catalyst during the long-term reaction could be shown in the Fig. 3j. Under the harsh reaction conditions, the ceria support sintered partially, but copper species were still anchored as clusters on the catalyst surface. However, for the reference 15CuAl catalyst, even though the $Al_2O_3$ support was very stable under high temperature, copper species have agglomerated significantly during the reaction, which caused severe deactivation in activity (Supplementary Fig. 13). And the 15CuCe catalyst prepared by the impregnation (IMP) method with poor dispersion of copper also exhibited much inferior activity (Supplementary Fig. 14).

In Table 2, the actual Cu content of the copper-ceria catalysts was similar to that of the theoretical value. And the $Cu/CeO_2$ catalysts exhibited similar specific BET surface areas (80.1–88.2 m²·g⁻¹) except the 15CuAl catalyst showed a larger value 154.7 m²·g⁻¹. X-ray diffraction (XRD) results (Supplementary Fig. 15) showed that fluorite $CeO_2$ served as dominate phase for all the fresh and used catalysts except the 25CuCe catalyst. And from the XRD results

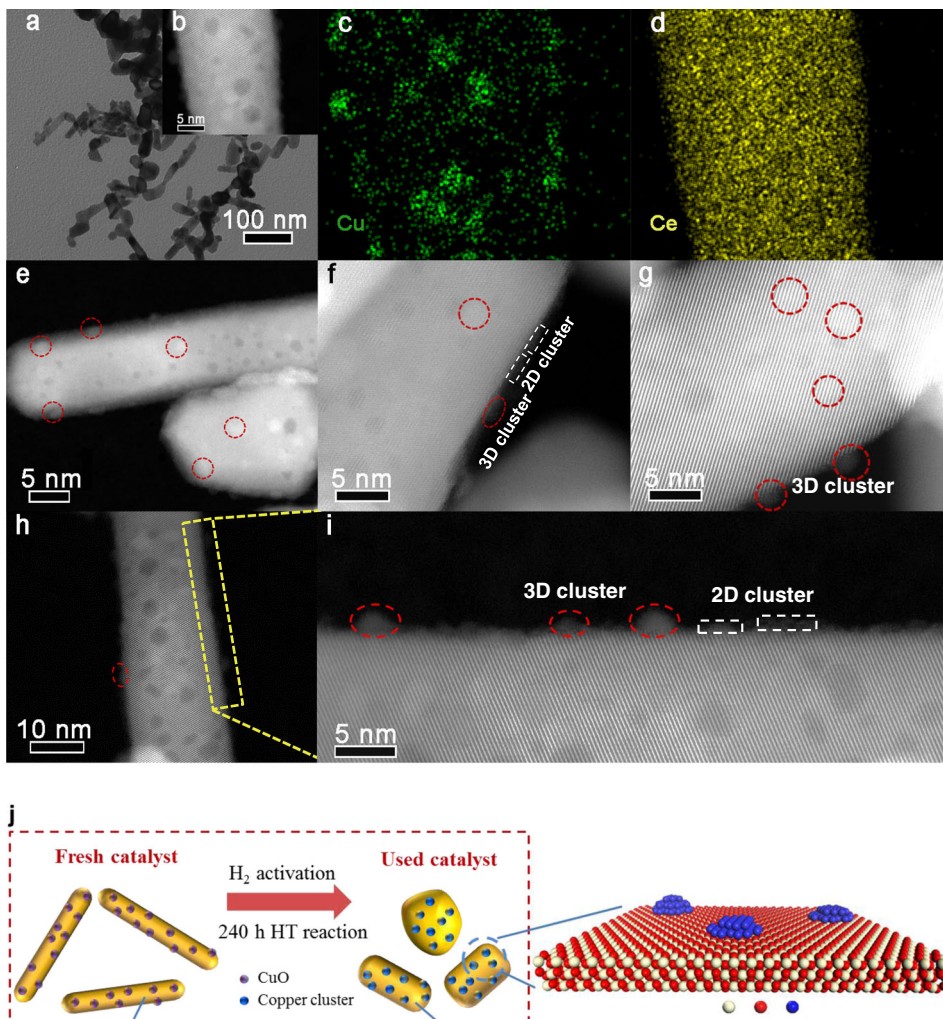

**Fig. 3 Structure characterization of the 15CuCe catalyst after 240 h stability test. a** TEM image; **b–d** STEM images and element mapping results. **e–h** HAADF-STEM images; **i** Enlarged image of the highlighted region in (**h**); **j** Scheme of structural evolution for the copper–ceria catalyst during long-term stability test.

**Table 2 Physicochemical properties of catalysts.**

| Catalyst | Cu (wt.%)[a] | $S_{BET}$ (m$^2$·g$^{-1}$)[b] | $H_2$ ($\mu$mol·g$^{-1}$)[c] | Integral $D/F_{2g}$ signal[d] |
|---|---|---|---|---|
| 1CuCe | 0.8 | 86.2 | $\alpha$430, $\beta$124 | $\alpha$0.11, $\beta$0.16 |
| 5CuCe | 3.4 | 88.2 | $\alpha$1191, $\beta$510 | $\alpha$0.47, $\beta$0.67 |
| 10CuCe | 9.7 | 83.9 | $\alpha$2543, $\beta$1352 | $\alpha$0.60, $\beta$0.96 |
| 15CuCe | 15.2 | 83.3 | $\alpha$3791, $\beta$1996 | $\alpha$0.65, $\beta$1.50 |
| 25CuCe | 26.5 | 80.1 | $\alpha$4647, $\beta$3062 | $\alpha$0.75, $\beta$1.69 |
| 15CuAl | 16.1 | 154.7 | / | / |

[a]Determined with ICP-OES.
[b]Determined with $N_2$ adsorption.
[c]Actual values of $H_2$ consumption ($\alpha$) and theoretic values of $H_2$ consumption calculated according to $Cu^{2+} \rightarrow Cu^0$ ($\beta$).
[d]$D/F_{2g}$ internal ratio of fresh catalysts ($\alpha$) and used catalysts ($\beta$).

(Supplementary Fig. 13) of the fresh 15CuCe and used 15CuCe catalysts after 70 h stability test, it could be seen that copper species existed mainly in the form of CuO before the reaction and metallic $Cu^0$ after the reaction. The tiny diffraction peak of $Cu^0$ also suggested that copper could maintain very small size even after a long period of high-temperature reaction, which was also consistent with the results of STEM. The Cu 2$p$ X-ray photoelectron spectroscopy (XPS) spectra of the fresh, activated, and used 15CuCe catalyst were shown in Supplementary Fig. 16. The XPS peaks centered at 933.6 and 932.4 eV were attributed to the Cu 2$p_{3/2}$

region; According to the previous reports, the peak centered at 933.6 eV was attributed to the $Cu^{2+}$ species, and 932.4 eV was assigned to $Cu^+/Cu^0$ species[30]. It indicated that only $Cu^{2+}$ was detected for the fresh 15CuCe catalyst, while for the activated and used 15CuCe sample, the $Cu^+/Cu^0$ species appeared. This result indicated that the surface $Cu^{2+}$ species was reduced to $Cu^+$ or $Cu^0$ in the activation process.

**The interaction between the copper and ceria in the catalyst.** The high dispersion of the copper species reflected the interaction between copper and ceria. For the fresh $Cu/CeO_2$ catalysts, such interaction can be confirmed by $H_2$ temperature-programmed reduction ($H_2$-TPR)[8]. As shown in supplementary Fig. 17a, the TPR profiles could be deconvoluted into three peaks. In the previous report[31], the EXAFS data confirmed the existence of Cu-O and Cu-Ce binding in the $Cu/CeO_2$-NR catalyst, which was quite consistent with the reduction peaks of $CuO_x$ clusters and the Cu-[$O_x$]-Ce structure. However, the experimental result in the Supplementary Fig. 17b and other reports[32] indicated that the $H_2$-TPR pattern of pure CuO was not completely symmetric, which suggested that the reduction of $CuO_x$ was not completed in one step, suggesting the $CuO_x$ species were progressively reduced to $Cu^+$ and $Cu^0$ species. Besides, the reduction peak of CuO

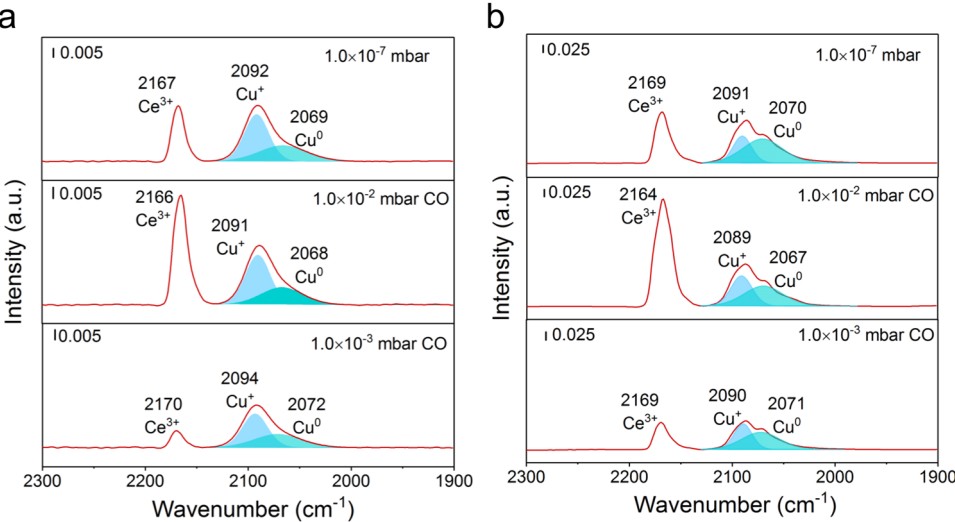

**Fig. 4 The interaction between copper and ceria. a–b** In situ infrared spectra recorded after exposing the 15CuCe catalyst to CO with different partially pressure at –143 °C after $H_2$ pretreatment (**a**) and 70 h stability test at 600 °C (**b**), respectively.

could not be deconvoluted into two symmetric peaks, indicated the reduction of $Cu^{2+}$ to $Cu^+$ and the reduction of $Cu^+$ to $Cu^0$ occurred simultaneously in a certain range of reduction temperature. Therefore, we speculated the reduction peaks of highly dispersed $CuO_x$ clusters in copper–ceria catalysts were also not symmetric, and the α and β peaks could not be attributed to the reduction of single species, but the progressively reduction of $CuO_x$ species to $Cu^+$ and $Cu^0$ species. Furthermore, the high-temperature sharp peak (170–260 °C) was due to the reduction of the strong interaction of Cu-$[O_x]$-Ce structure[30]. Comparing to the $H_2$-TPR results of pure $CeO_2$ and CuO (Supplementary Fig. 17b), the CuO–$CeO_2$ interaction enhanced the redox properties of these catalysts[33]. However, the reduction temperature of the reference 15CuAl catalyst was close to that of pure CuO, which meant there was a very weak interaction between copper and alumina (Supplementary Fig. 18). With such weak interaction, copper were difficult to be stabilized on the $Al_2O_3$ support under the high-temperature reduction conditions, causing the severe deactivation of the 15CuAl catalyst.

In order to further explore the interaction between the interfacial copper and ceria in the 15CuCe catalyst after the $H_2$ pretreatment and the stability test, the in situ infrared spectroscopy at the low temperature (−143 °C) was measured by using the CO as probe molecule. As shown in Fig. 4a, three CO bands appeared after $1.0 \times 10^{-3}$ mbar CO was injected. The band at 2166–2170 cm$^{-1}$ was assigned to CO adsorbed at the $Ce^{3+}$ site[25]. And the two bands at 2092 cm$^{-1}$ and 2069 cm$^{-1}$ were attributed to the CO adsorption on the $Cu^+$ and $Cu^0$ sites[34], respectively. The intensity of all CO bands increased with the increase of CO pressure, especially at the $Ce^{3+}$ site. Then with the rising of the degree of vacuum to $1.0 \times 10^{-7}$ mbar, the rapid elimination of adsorbed CO at the $Ce^{3+}$ was observed, which was related to the weak binding energy of CO-$Ce^{3+}$. The CO-$Cu^+$ related infrared bands demonstrated that part of $Cu^+$ sites could not be reduced during the $H_2$ activation process. Besides, as for the partially sintered 15CuCe sample after the stability test, there were also three bands appeared (Fig. 4b). The presence of the $Cu^+$ site and the $Ce^{3+}$ site confirmed that the interaction between the positively charged copper atoms with eletrophilicity and the $Ce^{3+}$ with nucleophilicity was not destroyed after long time treatment of reductive atmosphere at high temperature of 600 °C. The $H_2$-TPR result for the 15CuCe catalyst after the temperature-programmed surface reaction (TPSR) test also gave a reduction

peak (Supplementary Fig. 19), which again suggested a part of copper species remain $Cu^{\delta+}$ with electrophilicity during the reaction due to the interaction between copper and ceria[32]. Therefore, there was no doubt that the stable interaction between copper and ceria were present, which ensured the high dispersion and high stability of the active copper sites under harsh conditions.

**Role of the surface oxygen vacancy in the catalysts.** Due to the interaction between copper and ceria, the $Ce^{4+}$ can be transformed to $Ce^{3+}$, which is accompanied by the formation of oxygen vacancy[8,34–36]. It has been reported that oxygen vacancies commonly are recognized as the crucial active site to adsorb and dissociate $CO_2$[28,37], playing an important role in the $CO_2$ reduction reaction. Luis F. Bobadilla et al. demonstrated that the dissociation paths of $CO_2$ on reductive and non-reductive supports were different, the oxygen vacancies on reductive supports could activate $CO_2$ more efficiently[38]. However, $CO_2$ is a stable molecule, whose dissociation rate is closely related to the number of oxygen vacancies[39]. In this work, ex situ and in situ Raman spectra were performed to characterize the oxygen vacancies of the catalysts. In ex situ Raman results of all fresh and used copper–ceria catalysts (Supplementary Fig. 20), besides the vibration mode ($F_{2g}$) of $CeO_2$ fluorite-type structure at ~454 cm$^{-1}$, a broad $D$ band was also found[8]. The $D_1$ peak located at ~543 cm$^{-1}$ resulted from surface oxygen vacancy where $Ce^{4+}$ was replaced by $Ce^{3+}$. And the $D_2$ peak at ~603 cm$^{-1}$ was the intrinsic defect in ceria[40,41]. The relative integral intensity ratio of $D/F_{2g}$ reflected the concentration of oxygen vacancies in each catalyst[3,8,42,43]. As illustrated in Supplementary Figure 20, the relative strength of the $D$ band gradually increased with the increase of copper loading which meant that copper could promote the creation of oxygen vacancies. And the oxygen vacancy concentration of the used catalysts was higher (Table 2), indicating that more oxygen vacancies were created during the pretreatment and reaction process. The $H_2$-TPR results of all the Cu/$CeO_2$ catalysts also reflected that the amounts of hydrogen consumption of these catalysts were bigger than the theoretical values based on the complete reduction of $Cu^{2+}$ to $Cu^0$ (Table 2). This was due to the reduction of surface oxygen of the ceria at relatively low temperature by the aids of highly dispersed copper clusters. The more practical $H_2$ consumption exceeded the theoretical value, the more

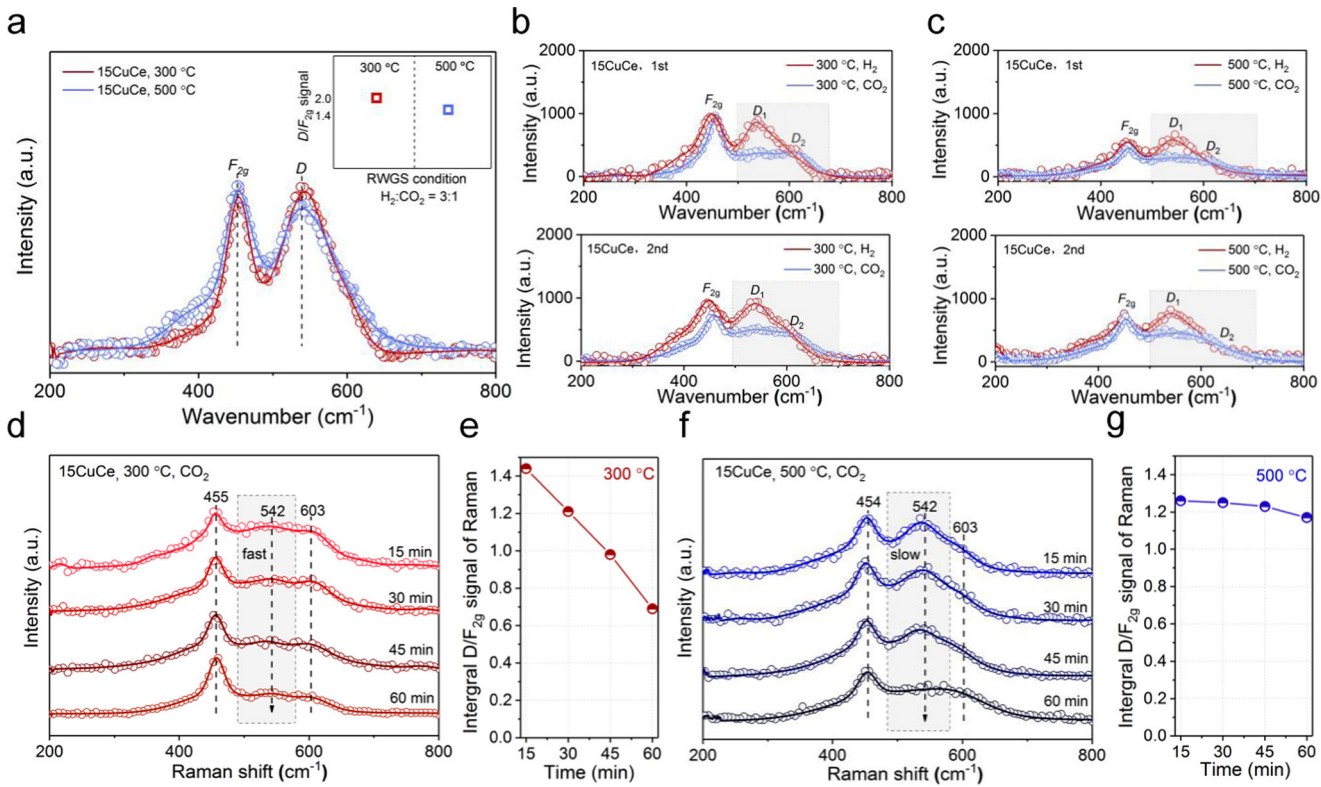

**Fig. 5 Examination of oxygen vacancies in the copper–ceria catalysts. a** In situ Raman under the RWGS reaction conditions for the 15CuCe catalyst; **b–c** In situ Raman of the 15CuCe catalyst with $H_2/CO_2$ switching under 300 °C and 500 °C, respectively; **d–e** The in situ Raman spectra over the 15CuCe catalyst under $CO_2$ flow at 300 °C and the variation of $I_D/I_{F2g}$ intensity ratio with time; **f–g** The in situ Raman spectra over the 15CuCe catalyst under $CO_2$ flow at 500 °C and the variation of $I_D/I_{F2g}$ intensity ratio with time.

oxygen vacancies were formed[44]. And for the $Cu/CeO_2$ catalysts after reaction, more oxygen vacancies were produced on the surface by the effect of the reductive reaction gas (69% $H_2$/23% $CO_2$/ 8% $N_2$).

To explore the role of oxygen vacancy in the actual reaction process further, the in situ Raman under reaction condition at 300 °C and 500 °C were measured and the results were showed in Fig. 5a. The 15CuCe catalyst showed a strong peak centered at 543 cm$^{-1}$, which was assigned as surface oxygen vacancies. Compared to the fresh 15CuCe catalyst, the $D_1$ band was even pronounced than $F_{2g}$ peak during the reaction process, which meant that abundant surface oxygen vacancies were in situ generated during the reaction process. For the 5CuCe catalyst, the $D_1$ band was also obvious, which meant surface oxygen vacancies were ubiquitous in the reaction process for $Cu/CeO_2$ catalysts in this work (Supplementary Fig. 21). And as exhibited by the illustrations in Fig. 4a and Supplementary Fig. 21, the in situ Raman tests reflected that the copper-ceria catalyst contained similar concentrations of oxygen vacancies at 300 °C and 500 °C. The small difference in the concentration of the oxygen vacancy could be considered within the error range. Besides, for the pure $CeO_2$ support (Supplementary Fig. 22), the intensity of the $D$ peak was very weak in the reaction atmosphere, which again demonstrated that the addition of copper could create more oxygen vacancies on the surface of ceria.

In order to explore the relationship between the activation of reactant molecules ($H_2$ and $CO_2$) and the oxygen vacancy, the in situ Raman spectra of the 15CuCe catalyst with $CO_2/H_2$ switch under 300 °C and 500 °C were also measured. As shown in the Fig. 5b and c, the Raman spectra of the 15CuCe catalyst after $H_2$ treatment showed a strong characteristic peak of oxygen surface vacancy centered at 543 cm$^{-1}$. When $CO_2$ was filled in, the

intensity of $D_1$ peak diminished, which meant that $CO_2$ adsorbed on the surface oxygen vacancies and occupied them. However, the stable $D_2$ peak suggested that $CO_2$ adsorption almost has little effect on the intrinsic defects. This experimental phenomenon directly confirmed that the surface oxygen vacancy was involved in this reaction. The second cycle of in situ Raman measurement also gave the same conclusion that surface oxygen vacancies were created by $H_2$, and consumed by $CO_2$. This in situ Raman result indicated that the surface oxygen vacancies could be consumed and regenerated as the reaction progress, allowing for the existence of lots of surface oxygen vacancies on the catalyst surface which could promote the high activity and stability. Besides, the relationship between the concentration of oxygen vacancy and $CO_2$ treatment time at 300 °C and 500 °C was also explored. As shown in the Fig. 5d–g, the decrease rate of oxygen vacancy at 300 °C was significantly faster than that at 500 °C. Combined with the results of the $CO_2$-TPD in Supplementary Fig. 23, the adsorption of $CO_2$ became more and more difficult with the increase of temperature. Therefore, it was easier for $CO_2$ to adsorb on oxygen vacancies at 300 °C compared to 500 °C. Besides, the effects of $H_2$ and $CO_2$ concentrations on the oxygen vacancy have also been investigated. As shown in Supplementary Fig. 24, the addition of $CO_2$ could reduce the amount of oxygen vacancies. However, further increasing the concentration of $CO_2$ did not significantly reduce the concentration of oxygen vacancies, which suggested that the rate. which $CO_2$ consume oxygen vacancies was much slower than the rate of which $H_2$ produced oxygen vacancies. The rapid formation of oxygen vacancies also meant the 15CuCe catalyst with dispersed copper clusters has a strong ability to dissociate $H_2$.

Meanwhile, as illustrated in Supplementary Fig. 25, the reaction orders of $CO_2$ for the 15CuCe and 5CuCe catalysts

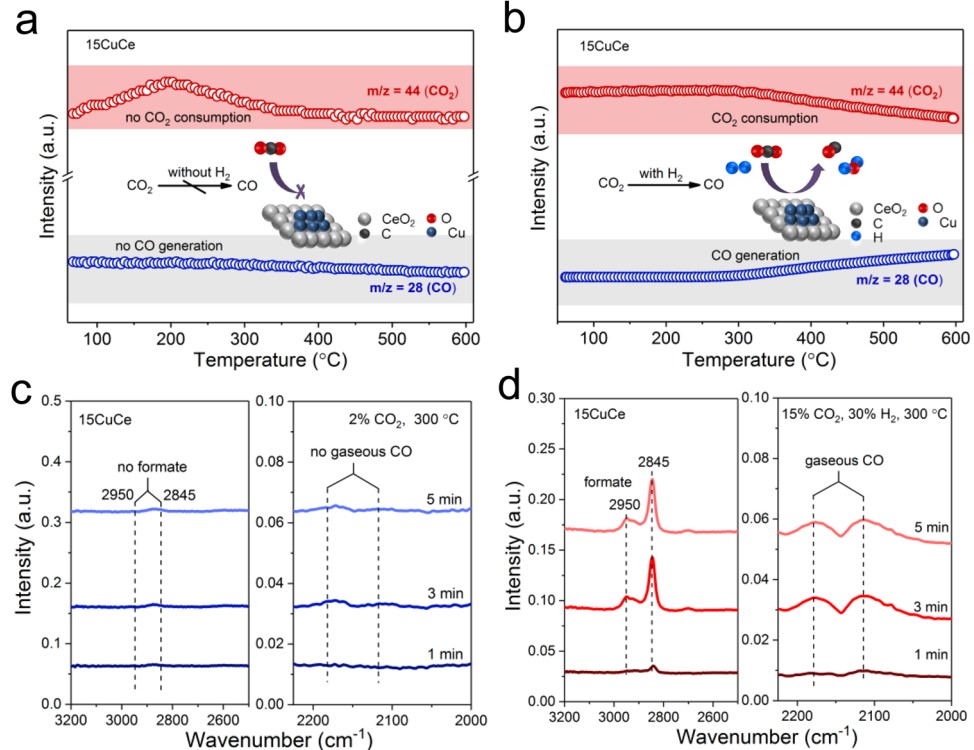

**Fig. 6 RWGS mechanism and reactive intermediates study of the 15CuCe catalyst. a** The $CO_2$ dissociation experiment of the 15CuCe catalyst; **b** TPSR results of the 15CuCe catalyst; **c–d** In situ diffused reflectance infrared Fourier transform spectroscopy (DRIFTS) spectra of 15CuCe catalyst during $CO_2$ treatment and reaction conditions at 300 °C, respectively.

were 0.25 and 0.52, respectively. And the reaction orders of $H_2$ over the 15CuCe and 5CuCe catalysts were 0.25 and 0.3, respectively. The lower apparent reaction orders of $CO_2$ and $H_2$ on the 15CuCe catalyst compared to the 5CuCe sample reflected the reaction rate on the 15CuCe catalyst was less dependent on the concentrations of $CO_2$ and $H_2$, which might suggested $CO_2$ and $H_2$ were relatively easily adsorbed on the 15CuCe catalyst with more oxygen vacancies and copper sites.

**Reaction mechanism study.** The mechanisms of the RWGS reaction have been classified into two categories, redox mechanism. and associative mechanism[19]. Whether the dissociated H species involve in the formation of reactive intermediates (such as formate) is the key to distinguish these two mechanisms[45]. In this work, the dissociation experiment of $CO_2$ was performed to probe the reaction pathway. After the catalyst pretreated by $H_2$/Ar at 600 °C for 1 h, the $CO_2$/Ar mixed gas was introduced into the reactor at room temperature. As shown in Fig. 6a, there was no consumption of $CO_2$ except for a part of the desorbed $CO_2$ during the adsorption process. Meanwhile, there was no generation of CO could be found. The above experimental results indicated that it was difficult for $CO_2$ itself to be directly dissociated to form CO by the 15CuCe catalyst. In Fig. 6b, the TPSR result illustrated that $CO_2$ signal gradually decreased and CO signal gradually increased from ~300 °C, suggesting $CO_2$ was converted into CO with the assistance of $H_2$. Thus, combing the results of $CO_2$ dissociation experiment and TPSR, it could be concluded that $CO_2$ activation may processed via an associative intermediate pathway.

To further explore the active intermediates, the in situ diffused reflectance infrared Fourier transform spectroscopy (DRIFTS) was carried out. As shown in the Fig. 6c and Supplementary Fig. 26a, after the injection of $CO_2$, only carbonate signal appeared and no CO gas signal was generated, which indicated

carbonate was hard to dissociate directly into CO. However, as shown in Fig. 6d and Supplementary Fig. 26b, after the injection of $CO_2$ and $H_2$ over the activated 15CuCe catalyst, in addition to the carbonate signal, the C=O vibration peak of formate at 1373 $cm^{-1}$ and the typical C-H stretch vibration peaks of formate at 2949 $cm^{-1}$ and 2845 $cm^{-1}$ were observed[46,47]. Simultaneously, the broadband, which was attributed to the gaseous CO at 2000–2200 $cm^{-1}$ has been detected as the increasing of the formate signal.

Calculations based on density functional theory (DFT) were also performed. The adsorption characteristics of $CO_2$ suggested that the presence of Cu atoms assured that the decreasing entropy step could occur, shown as Fig. 7a, b and Supplementary Table 2. Notwithstanding the situation of oxygen vacancy affected the binding force, the energy could be reduced more than one electron-volt under the bonding interaction between Cu and $CO_2$. And in the presence of $H_2$, Cu atoms captured $H_2$ molecule and broken H-H bond, then transferred H atom to $CO_2$, shown as Fig. 7c. Formate structures formed accompanying the formation of C-H bond, and these structures exhibited in the intermediate IMA3, IMA3-I, IMA4, and IMA4-I. The heat liberation declared that the formate formation was a thermodynamic feasible elementary reaction. The subsequent hydrogen-migration step ($\Delta E = 1.587$ eV) was the thermodynamically limiting step, and this might be the reason that formate signals were detected by DRIFTS. Carboxylic intermediates, IMA3-II and IMA4-II, were involved in the mechanism at the same time. Different from the formate intermediates, one step was absent from the carboxylic path, i.e., IMA4-II produced IMA6 directly. Based on the above experimental results and DFT calculations, we could further infer that the associative mechanism was involved in this reaction and the surface formate and carboxylic species might be the important reactive intermediates. The synergistic catalytic effect between copper clusters and oxygen vacancies copper clusters promoted

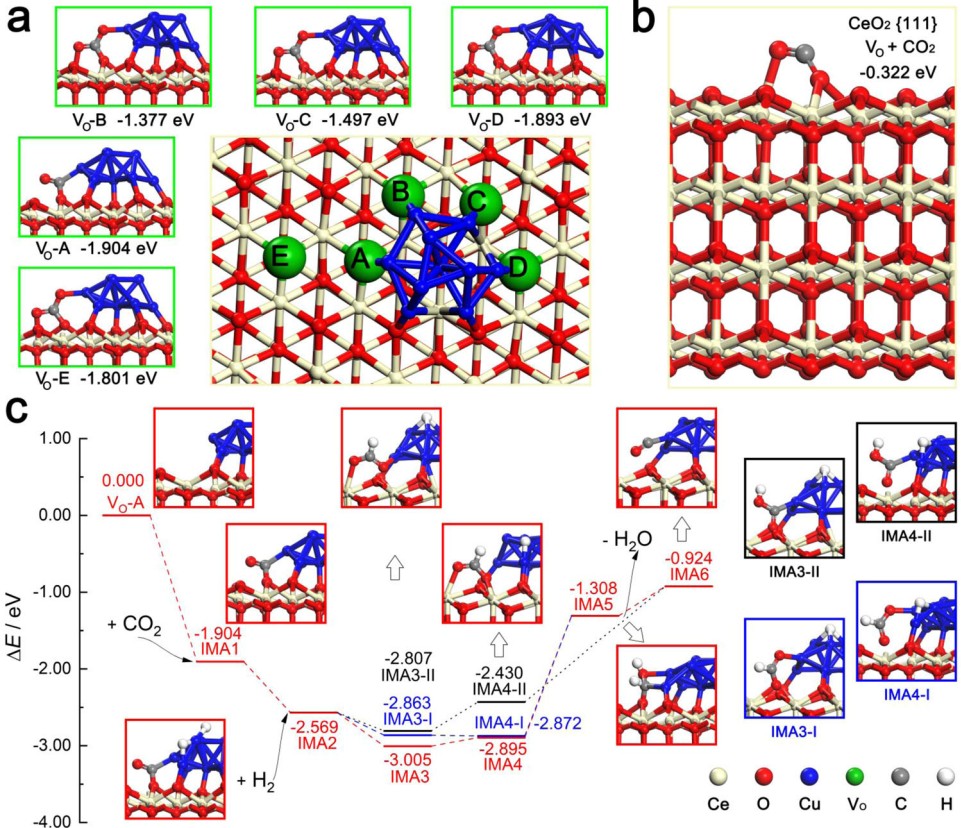

**Fig. 7 The proposed reaction pathways for the RWGS reaction on the copper–ceria catalyst. a** Chemisorption of $CO_2$ on the 10Cu/CeO$_2${111} surface. Five oxygen vacancies, named after $V_O$-A to $V_O$-E, were made comparisons, and the selected $CO_2$ was located close to $V_O$-A; **b** The adsorption energy that $CO_2$ was bound to $V_O$ on the CeO$_2${111} surface was obviously weaker than those of Cu dropped ceria sites; **c** RWGS reaction mechanism occurred in the $V_O$-A. The red, blue, and black lines indicated different reaction paths, and the structural diagrams with the red, blue or black stroke corresponded to the reaction paths, respectively.

the adsorption of $CO_2$ and the formation of active intermediates. The sufficient copper clusters and abundant oxygen vacancies in the 15CuCe catalyst undoubtedly created more metal cluster-oxygen vacancy active interfaces.

## Discussion

For high-temperature catalytic reactions, the development of catalysts with both high activity and excellent stability has always been difficult. In this work, highly dispersed active copper clusters with high loading (15 wt.%) were stably constructed on the partially sintered copper–ceria catalyst during the real reaction process. The optimal 15CuCe catalyst exhibited excellent catalytic performance to catalyze the RWGS reaction at high operating temperature, which surpassed almost all the reported non-noble metal catalysts and costly noble metal catalysts. The harsh reaction conditions of high temperature and reductive atmosphere caused the ceria support sintered partially, while the interaction between copper and ceria maintained well. The unexpected stable interaction ensured the copper species to maintain stable in the forms of 2D layered clusters and 3D hemisphere-shaped clusters on the partially sintered ceria support. Besides, abundant surface oxygen vacancies were in situ generated and consumed circularly during the reaction process, forming the synergistic catalytic effect with copper cluster to promote the activation of $CO_2$ and the formation of active intermediates. The unmatched activity and solid stability of this catalyst show great potentials in the practical applications. And the reveal of the structure-function relationship of the catalyst with sintered configuration also provides a reference for other systems.

## Methods

**Preparation of copper–ceria catalysts.** The copper–ceria catalyst was prepared by the deposition-precipitation (DP) method. Firstly, the ceria support (0.50 g) was dispersed in 30 mL high purity water under continuous stirring. Next, different amounts of copper precursor, Cu(NO$_3$)·3H$_2$O, were dissolved in 12.5 mL of high pure water, and then added into the above CeO$_2$/H$_2$O suspension dropwise. During the process of instilling, the pH value of the solution was controlled to ca. 9 by adding Na$_2$CO$_3$ solution (0.50 mol·L$^{-1}$). The obtained precipitates were further aged at room temperature for 1 h before filtration, followed by washing with high pure water (1 L) at room temperature. The resulting material was dried in air at 75 °C overnight and then calcined in still air at 600 °C for 4 h (heating rate: 2 °C/min). The copper–ceria samples synthesized in this work were donated as $x$CuCe ($x$ = 1, 5, 10, 15 and 25), where $x$ is the copper content in weight percent ($x$ = [Cu/CeO$_2$]wt × 100%). The reference copper–ceria catalyst was prepared using the impregnation (IMP) method. 0.5 g of CeO$_2$ support was dispersed in deionized water by stirring. Then the suitable amount of copper nitride was added into the slurry. The obtained mixture was dried at 90 °C using an oil bath under stirring. The resulting material was calcined in still air at 600 °C for 4 h (heating rate: 2 °C/min). The reference copper–ceria sample was donated as 15CuCe-IMP.

**Transmission electron microscopy (TEM).** Transmission electron microscopy (TEM) was undertaken by JEM-2100F (JEOL) instrument operating at 200 kV. The samples were dispersed in ethanol by ultrasonic and dropped on the carbon-coated Cu grid before test. The images of high-resolution TEM (HR-TEM) were obtained by using a JEOL JEM-2800 instrument with an acceleration voltage of 200 kV. The element mapping results and EDS analysis were acquired from the same machine under STEM mode. The High-angle annular dark-field scanning transmission electron microscopy (HAADF-STEM) images were obtained on a Thermo Scientific Themis Z microscope equipped with a probe-forming spherical-aberration corrector.

**Raman test.** All the ex situ and in situ Raman spectra were acquired by using a Raman microscope system (HORIBA JY) with laser excitation at 633 nm. The integration times of ex situ and in situ Raman spectra were 1 min and 5 min, respectively.

**In situ infrared spectroscopy in the transmission mode.** The infrared measurements were conducted in a UHV apparatus combining a FTIR spectrometer (Bruker Vertex 70 v) with a multi-chamber UHV sytem. The sample was pretreated with $H_2$ or reaction gas at 873 K for 30 min, and then exposed to CO with desired pressure at 130 K.

**Catalytic tests and kinetics measurement.** The catalytic performance evaluation was tested in a fixed-bed flow reactor under a gas atmosphere of 23% $CO_2$/69% $H_2$/ $N_2$ (66.7 mL·min$^{-1}$, Deyang Gas Company, Jinan) at 1 bar total pressure. Before the activity test, 10 mg catalysts (40-60 mesh) diluted with 90 mg inert $SiO_2$ were activated by 5% $H_2$/Ar at 600 °C for 60 min followed by switching to the feed gas for testing. The test temperature ranges from 300 °C to 600 °C. And before the analysis of gas products, the RWGS reaction needs to stabilize for 60 min at each test temperature. The gas products were analyzed by using an on-line gas chromatograph equipped with a thermal conductivity detector (TCD). $CO_2$ conversion and CO selectivity were calculated using the following equations:

$$X_{CO_2} = \frac{n_{CO_2}^{in} - n_{CO_2}^{out}}{n_{CO_2}^{in}} \times 100 \qquad (1)$$

$$S_{CO} = \frac{n_{CO}^{out}}{n_{CO}^{out} + n_{CH_4}^{out}} \times 100 \qquad (2)$$

where $n_{CO_2}^{in}$ is the concentration of $CO_2$ in the reaction stream, and $n_{CO_2}^{out}$, $n_{CO}^{out}$, $n_{CH_4}^{out}$ are the concentrations of CO, $CO_2$, $CH_4$ in the outlet. The $CO_2$ thermodynamic equilibrium conversion was calculated from HSC chemistry software version 6.0. The ratio of $H_2$:$CO_2$ in the initial state was 3:1, and the products including CO, $H_2O$ and $CH_4$ were taken into account in the calculation process. For all catalysts, the $E_a$ was measured by using the same reactor for catalytic performance above. Appropriate amount of catalysts diluted with inlet $SiO_2$ were used in the kinetics experiments. And in order to obtain accurate kinetics data, the catalysts need to be first treated with reactive gas for an hour at 600 °C. During the kinetic test, the $CO_2$ conversion remained between 5% and 15% by changing gas flow rate. The reaction orders of $CO_2$ and $H_2$ for the catalysts were measured under 500 °C. The RWGS activity was recorded while the concentration of $CO_2$ or $H_2$ in the reaction gas was varied on purpose.

## Data availability
The main data supporting the findings of this study are available within the article and its Supplementary information. All other relevant source data are available from the corresponding author upon reasonable request. Source data are provided with this paper.

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

## Acknowledgements

This work was financially supported from the National Science Foundation of China (Grant No. 21771117, 21805167, 22075166), the Taishan Scholar Project of Shandong Province of China, the Young Scholars Program of Shandong University (Grant No. 11190089964158), the Outstanding Youth Scientist Foundation of Hunan Province (Grant No. 2020JJ2001), and the Key Project of Educational Department of Anhui Province (Grant No. KJ2019A0861). We thank the Center of Structural Characterizations and Property Measurements at Shandong University for the help on sample characterizations.

## Author contributions

C.J.J. and C.M. supervised the work; H.X.L. and C.J.J. designed the experiments, analyzed the results and wrote the manuscript; S.Q.L. made the DFT calculations; H.X.L. and W.W.W. performed the in situ DRIFTS, in situ Raman; H.X.L. and W.Z.Y. performed the catalysts preparation, catalytic tests and the TPR tests; W.J.Z. and C.M. performed the aberration-corrected HAADF-STEM measurements and analyzed the results.

## Competing interests

The authors declare no competing interests.
