## [Peer Review File · Nature Communications]

Title: Partially sintered copper–ceria as excellent catalyst for the high-temperature reverse water gas shift reactionREVIEWER COMMENTS

Reviewer #1 (Remarks to the Author):

This paper presents an interesting structural/chemistry study of Cu/Ce based catalyst using the RWGS reaction as potential application. Overall, the manuscript is well-written and provides valuable characterization data which can be helpful for researchers working in this field.

Nevertheless, from the catalysis angle, the paper does not present substantial experiments, discussion, and innovation. It is worthily to mention that the conditions used are very harsh and challenging.

However, there are no enough catalysis tests on the manuscript and overall the catalytic aspects are vaguely analysed.

On the other hand, the characterization presented is very complete and interesting but does not reveal anything new that was not already known.

Hence due to the lack of mandatory test and solid and reliable contributions to the catalysis discipline, this paper is not within the top papers catalysis field and does not level up the high standards of Nature Communication right now. There is a lack of basic catalytic tests that are essential to be able to affirm that CuCe sample is an excellent catalyst for high-temperature rWGS.

Some comment that should to be address are given below.

- Please provide information about how the authors have calculated the equilibrium curve. Comparing this presented curve with other works in literature, at low reactions T in similar conditions the equilibrium conversion should be higher since methanation reaction is present.
- Please explain the reason why the author choose a H₂:CO₂ ratio of 3:1 and not 4:1 or others. How their catalysts works in others H₂:CO₂ ratios.
- Focusing practical application (as authors mention in the manuscript), a stability test of 70 h is not enough challenging. To prove the stability of this catalysts (Cu and ceria agglomeration and deactivation) more demanding tests has to be consider of a minimum of 1 week.
- What about the recyclability of this catalyst. This sample suffer a remarkable ceria agglomeration/collapse.
- What about the reproducibility of this catalyst. Both from the point of view of the synthesis and the reaction.
- Table 1 compares mostly Cu-based catalyst. Other transition metal catalyst should be compared in this table (for example Fe-based catalysts and more Ni-based catalyst).
- Regarding characterization, reduced samples' characterization should be mandatory in order to know exactly the morphology/chemistry of your sample before the reaction. It is more interesting to know the

state of your active phases. Furthermore, during the text can be seen several affirmations about Cu being reduced during the reaction, but this statement is not correct since Cu was already reduced in the pre-activation treatment (described in the section “method”)

- Homogeneity in the T units, no a mixture of °C and K (IR).

- Synthesis. In the CeO₂ preparation procedure the solid obtained was not calcined until after the Cu impregnation, however, the alumina was calcinated previous to Cu deposition. This can bring differences in the support – Cu interaction.

Reviewer #2 (Remarks to the Author):

The work submitted by Prof. Jia, Prof. Ma and coworkers is an excellent study of rWGS reaction using an economical, active and stable Cu/CeO₂ catalyst. The results obtained are interesting and the mechanistic study proposed provides some relevant insights. However, there is not novelty and the impact is far from the objective of Nature Communications. Therefore, I recommend to submit this work to other journal specialized in catalysis such as J. Catal. or ACS Catal.

Reviewer #3 (Remarks to the Author):

Key results:

In this research study, Liu et al. have designed a Cu-based catalyst with very high activity and stability. The catalyst performs better than previously reported Cu-based catalysts for the RWGS reaction. They also showed the effect of various Cu loading, the role of oxygen vacancies in the reaction, and the SMSI effect on the catalyst activity.

Validity:

The data and results are valid and the data seem to be collected and interpreted in an acceptable way based on the description of the experiments.

Significance:

The findings are important due to the high activity and stability of the proposed catalyst for the reverse water gas shift reaction. However, Cu-ceria is an obvious system to prepare and test for scientists in this community. We have prepared copper on standard ceria support (no nanorods!) by wet impregnation and the activity was clearly below that of Cu-alumina. The main question why this specific catalyst is so much more active than other catalysts, particularly other Cu-ceria catalysts, has not been addressed by the authors, which limits the value of the paper for the scientific community. It is not clear what the reader should learn about the preparation of RWGS catalysts, besides that this very special catalyst is very active. The described material characterization data is solid and clear but does not reveal any

principally new insight on this catalyst type, which one would expect from a publication in Nature Communications. Therefore, I strongly recommend the authors to conduct additional experiments to elaborate the difference between their new catalyst and other RWGS catalysts, particularly standard Cu-ceria catalysts with the same composition.

Data and Methodology:

While the findings are interesting, I think some of the arguments are not at the level expected for publication at Nature Communication. Below are some of the points for the authors to consider:

1- The authors indicated that they used CeO₂ nanorods but these nanorods deformed/transformed through the reaction. Why using nanorods are important if the morphology changes during the reaction? what happens if we use other morphologies of CeO₂? How does the new morphology affect the reaction? Do we know the surface area of the used catalyst? Does the surface area increase or decrease after the change of morphology? Why does the size of the CeO₂ crystals, counterintuitively, reduce after 70 h of reaction (based on the XRD patterns in Figure S7c)?

2- CO Selectivity is defined as the CO concentration over the total concentration of CH₄ and CO leaving the reactor. This either should be changed to $n_{CO}/(n_{CO_{in}} + n_{CO_{out}})$ to account for all possible products of the reaction (which might not have been detected), or there must be a carbon balance accompanied by TPO to make sure the system's carbon balance is valid.

3- Through Raman spectroscopy, it is shown that CO₂ is adsorbed on the oxygen vacancies. But no data on how CO₂ is adsorbed is presented. In-situ DRIFTS for CO₂ adsorption is needed so that we can see how CO₂ is adsorbed. The fact that CO₂ is adsorbed on the oxygen vacancies does not rule out the other possible active sites for CO₂ adsorption.

Analytical Approach:

1- The reaction mechanism study is weak. It clearly shows that the redox mechanism is not happening. But the sole detection of formate on the surface is not enough to conclude about the reaction pathway. There can be various CO formation routes such as the carbonate route and hydroxycarbonyl route¹. Formate can also be a spectator (or minor reaction intermediates) on the surface.^[1-3] The full spectra should be visible to see the formed species. Better methodologies (such as addition of DFT study, Modulation Excitation Spectroscopy, or other in-situ methodologies) should be used to prove the reaction mechanism.

2- There are more than two peaks in the H₂-TPR profiles of the catalysts. These peaks need to be identified and described.

3- Why do we observe a weaker signal for XPS after reaction (Figure S9)? Does Cu diffuse into

CeO₂? What is the state of the signal for the H₂-treated catalyst before the reaction?

4- It is not clear why there are less oxygen vacancies created when testing at 500 °C as opposed to 300 °C (Figure 4a).

5- In the last paragraph on page 14, the authors indicate that the lower apparent reaction order of CO₂ and H₂ on 15CuCe compared to 5CuCe shows the higher ability of this catalyst in adsorption and activation of CO₂ and H₂. This conclusion is not clear to me. What we can understand from the lower activation of H₂ and CO₂ on 15CuCe is that the reaction rate on this catalyst is less dependent on the concentration of CO₂ and H₂. This does not lead to the conclusion that CO₂ and H₂ can better be adsorbed and activated for the RWGS reaction. Stronger adsorption of CO₂ or H₂ does not necessarily lead to their higher activity or higher level of participation in the reaction. The authors need to clarify their statements and/or add supporting arguments.

Suggested Improvements:

Most suggested improvements are already listed above. However, some additional suggestions are shown below:

1- another test with a catalyst with Cu loading between 15 and 25 might be needed to see if 15% Cu is the optimum loading.

2- In Figure 4a and Figure S14, the insets should be explained.

3- The text need full revision. There are grammatical errors (“strong interacted... structure”, “the reductive support with oxygen vacancies was easier to activate CO₂”, first three lines of page 12, “the D1 band was even obvious than F2g peak during...”, “have no influence to the intrinsic defects”, etc.) as well as typos (Figure 3 caption, etc.) which need to be fixed.

Clarity and Context:

The manuscript is written clearly and it is understandable for the scientific community. Other than some grammatical errors and typos which were referred to in the previous comments, the text, in general, is well written.

References:

- [1] L.F. Bobadilla, J.L. Santos, S. Ivanova, J.A. Odriozola, A. Urakawa, ACS Catal. 8 (2018) 7455–7467.
- [2] A. Goguet, F.C. Meunier, D. Tibiletti, J.P. Breen, R. Burch, J. Phys. Chem. B. 108 (2004) 20240–20246.
- [3] M. Zhu, Q. Ge, X. Zhu, Transactions of Tianjin University. 26 (2020) 172–187.

Responses to the Reviewers' Comments and the Corresponding Revisions

To Reviewer 1:

General comment: This paper presents an interesting structural/chemistry study of Cu/Ce based catalyst using the RWGS reaction as potential application. Overall, the manuscript is well-written and provides valuable characterization data which can be helpful for researchers working in this field.

Nevertheless, from the catalysis angle, the paper does not present substantial experiments, discussion, and innovation. It is worthily to mention that the conditions used are very harsh and challenging. However, there are no enough catalysis tests on the manuscript and overall the catalytic aspects are vaguely analyzed.

On the other hand, the characterization presented is very complete and interesting but does not reveal anything new that was not already known.

Hence due to the lack of mandatory test and solid and reliable contributions to the catalysis discipline, this paper is not within the top papers catalysis field and does not level up the high standards of Nature Communication right now. There is a lack of basic catalytic tests that are essential to be able to affirm that CuCe sample is an excellent catalyst for high-temperature rWGS.

Response: Thanks for the reviewer's valuable comments and suggestions. First of all, as we all know, the conversion and utilization of CO₂ has become an important issue that the world has to face. The RWGS reaction, which can be regarded as a pivotal process of transitioning the CO₂ into valuable chemicals and hydrocarbon fuels, has received extensive attention from chemical researchers in recent years. The pursuing of the efficient catalysts with high activity and stability would never be out of date. **Although many excellent reports have systematically explored the structure-function relations of various catalysts** (W. Liu et al, *Nat. Catal.* **2020**, *3*, 411–417; J. Szanyi et al, *Angew. Chem. Int. Ed.* **2020**, *59*, 17657–17663; F. S. Xiao et al, *J. Am. Chem. Soc.* **2019**, *141*, 8482–8488, and so on), **the fabrication of catalysts**

with both high activity and high stability under harsh reaction conditions is still urgently to be developed.

Secondly, copper–ceria catalysts have been widely investigated because of their catalytic activity in many reactions. Recently, Shen et al. explored the interfacial structures in copper–ceria catalysts with low copper loading (~3 wt.%) after the treatments at different temperatures under H₂ gas flow for only 2 hours (W. J. Shen et al, *Nat. Catal.* **2019**, *2*, 334–341). And our group also revealed the structure of the coordination-unsaturated and atomically dispersed copper species in the copper–ceria catalyst after air-calcination at 800 °C (C. J. Jia et al, *J. Am. Chem. Soc.* **2019**, *141*, 17548–17557). **However, the structure of the copper–ceria catalyst, including the status of copper species and the morphology of ceria, is unknown under harsh reaction conditions (at high temperature and with reductive atmosphere), especially for the catalyst with relatively high copper loading.**

Finally, the unique redox properties (switch between Ce⁴⁺ and Ce³⁺) of CeO₂ enable the formation of oxygen vacancy within the crystal structure. The oxygen vacancy plays an important role in the reactions involving CeO₂-based catalysts. **Although many previous reports have pointed out that oxygen vacancy was related to the catalytic activity of CO₂ reduction (J. Szanyi et al, *JACS Au.* **2021**, *1*, 977–986), it is still unclear how the oxygen vacancy is involved in the reaction.**

For above existing scientific problems, we designed relevant experiments and carried out related researches. We think that it can meet the scope of *Nature Communications*. **We believe our findings will interest a broad readership in Catalysis and Materials Science according to the following aspects:**

1. We precisely determined the structure of the copper–ceria catalyst for the RWGS reaction under harsh reaction conditions.

The structure of the copper–ceria catalyst treated at high temperature (600 °C) and with reductive atmosphere was clearly revealed. **Under such a harsh reaction condition, the CeO₂ supports were only partially sintered, but not completely sintered.** With comprehensive characterizations, we found that **the interaction**

between copper and ceria maintained well after long-term catalytic tests (for 240 hours), which ensured the copper species with high loading (~15 wt. %) were still very stable on the partially sintered support in the forms of 2D and 3D clusters.

2. We directly explored the role of the oxygen vacancies in the RWGS reaction.

In situ Raman experiments were designed and carried out to investigate the role of oxygen vacancies. Based on the test results, we found that the surface oxygen vacancies could be created by H₂ and consumed by CO₂. **The *in situ* generated and consumed surface oxygen vacancies in the copper–ceria catalyst were directly confirmed to be involved in the RWGS reaction.**

3. We clearly revealed the synergistic effect between copper cluster and oxygen vacancy.

Based on the experimental results and DFT calculations, we found that **the surface oxygen vacancy had synergistic effect with the adjacent copper cluster, improving the activation of CO₂ and the formation of active intermediates.** The stable copper cluster-oxygen vacancy synergistic sites ensured the copper–ceria catalyst had unmatched activity to catalyze the RWGS reaction, which surpassing almost all other reported metal catalysts. Besides, this copper–ceria catalyst also maintained excellent stability in the 240 h activity evaluation under very harsh reaction conditions (600 °C, space velocity of 400,000 mL·g_{cat}⁻¹·h⁻¹).

After all, in this work, the efficient copper–ceria catalyst with both high activity and stability under very harsh reaction conditions has been constructed. And the structure-function relation of the partially sintered copper–ceria catalyst was clearly revealed. **We think these findings can provide a valuable reference for the exploration and application of sintered catalyst in the high temperature reaction.** We are looking forward to your next comments.

Comment 1: Please provide information about how the authors have calculated the equilibrium curve. Comparing this presented curve with other works in literature, at

low reactions T in similar conditions the equilibrium conversion should be higher since methanation reaction is present.

Response: Thanks for the reviewer's valuable comments. As shown in supplementary Figure 1b, the CO selectivity of all Cu/CeO₂ catalysts was 100% during the whole test process, and no methane was detected in the product. Thus, the calculation of the equilibrium curves was only related to the RWGS reaction. The thermodynamic equilibrium conversion of CO₂ in the RWGS reaction as a function of H₂:CO₂ inlet ratio at various temperatures was also given in the previous literature (Y. Q. Huang et al, *J. Energy Chem.* **2017**, 26, 854–867). In this work, the equilibrium curve was calculated by the HSC Chemistry software. Taking the CO₂ equilibrium conversion rate at 600 °C as an example, the specific calculation method is as follows.

$$\Delta_r G_m = \Delta_r H_m - T\Delta_r S_m = 6.98 \text{ kJ} \quad (1)$$

$$K_{eq} = e^{\frac{-\Delta_r G_m}{RT}} = 0.382 \quad (2)$$

$$K_{eq} = \frac{p_{H_2O} p_{CO}}{p_{CO_2} p_{H_2}} = \frac{\beta^2}{(3-\beta)(1-\beta)} = 0.382 \quad (3)$$

$$\alpha = \beta \times 100 = 60.3$$

Where K_{eq} is the equilibrium constant of the RWGS reaction at 600 °C, and α is the CO₂ equilibrium conversion rate at 600 °C. **The relevant supplements have been added in the revised manuscript on page 20, line 26–27 (highlighted in yellow).** Thanks for the reviewer's comment again.

Comment 2: Please explain the reason why the author choose a H₂:CO₂ ratio of 3:1 and not 4:1 or others. How their catalysts works in others H₂:CO₂ ratios.

Response: Thanks for the reviewer's valuable comments. High operating temperature and high concentration of reductive atmosphere often caused the severe deactivation of catalysts in the RWGS reaction. In order to explore the durability of the synthesized Cu/CeO₂ catalysts under harsh reaction conditions, the reaction gas containing high H₂ concentration with relative high ratio of H₂:CO₂ as 3:1 was chosen. According to the reviewer's comments, the catalytic performances of the 15CuCe

catalyst under other reaction atmospheres with H₂:CO₂ ratios of 2:1 and 4:1 were also evaluated. As shown in Figure R1, the CO₂ conversion increased with the increasing of the H₂:CO₂ inlet ratio at various reaction temperatures. And at all these three H₂:CO₂ ratios, the 15CuCe catalyst shown excellent catalytic performances, suggesting this catalyst had high catalytic efficiency over a wide range of H₂:CO₂ ratios. And it was noteworthy that even when the H₂:CO₂ ratio reached 4:1, no CH₄ was detected in the product, which indicated the catalyst catalyzed the RWGS reaction much rather than the methanation reaction. The corresponding description has been added in the revised manuscript on *page 5, line 18–20* and *page 6, line 1–6*. And the data was shown in supporting information as Figure S2 on *page S7* (highlighted in yellow).

Figure R1. Catalytic performances for the 15CuCe catalyst with H₂:CO₂ ratio of 2:1, 3:1 and 4:1.

Comment 3: *Focusing practical application (as authors mention in the manuscript), a stability test of 70 h is not enough challenging. To prove the stability of this catalyst (Cu and ceria agglomeration and deactivation) more demanding tests has to be consider of a minimum of 1 week.*

Response: The reviewer’s valuable comment was highly appreciated. Long-term stability was one of the most important indexes to evaluate the catalyst performance. To explore the durability of the 15CuCe catalyst further, a stability test was measured for 240 hours. As for the test result under the harsh condition (GHSV

= 400,000 mL·g_{cat}⁻¹·h⁻¹, 600 °C) in Figure R2, it maintained more than 85% of its initial activity after 240 h. During the first 20 hours of the stability test, the catalyst experienced slight deactivation, but the activity remained stable in the subsequent test. The result strongly confirmed the excellent stability of the 15CuCe catalyst in the high-temperature RWGS reaction. The corresponding data has been added as Figure 1d in the revised manuscript on page 5 (highlighted in yellow). Besides, the HAADF-STEM images of the used 15CuCe catalyst after the 240 h stability test (Figure R3) exhibited that CeO₂ nanorods were still partially sintered and the 2D and 3D copper clusters were still anchored on the partially sintered CeO₂ nanorods, which undoubtedly prevented the catalyst from deactivation. The corresponding data has been added as Figure 3 in the revised manuscript on page 9 and the corresponding discussion has been added in the revised manuscript on page 9, line 8–13 and page 10, line 1–11 (highlighted in yellow). Thanks for the reviewer's comment again.

Figure R2. Catalytic performance of the 15CuCe catalyst in the long-term RWGS reaction.

Figure R3. (a) TEM image of the used 15CuCe catalyst after 240 h stability test. (b–d) STEM images and element mapping (EDS) results of the used 15CuCe catalyst after 240 h stability. (e–h) HAADF-STEM images of the used 15CuCe catalyst after 240 h RWGS reaction. (i) Enlarged image of the highlighted region in (h).

Comment 4: What about the recyclability of this catalyst. This sample suffer a remarkable ceria agglomeration/collapse.

Response: Thanks for the reviewer’s valuable comments. The ceria nanorod support of the copper–ceria catalyst could suffer partial sintering during the high-temperature RWGS reaction. In order to explore the cyclic stability, the activity evaluation of six start-up cool-down cycles was carried out. And as illustrated in Figure R4, the reaction rate of the 15CuCe catalyst maintained stable in 6 rounds of activity tests, indicating the 15CuCe catalyst had very good stability under the start-up cool-down reaction conditions. It should be noted that the 15CuCe catalyst was treated by the RWGS reaction atmosphere at 600 °C for one hour before the kinetic tests. **The Figure R4 has been added as Figure S6 in the revised supporting information on**

page S11. And the corresponding description has been shown in page 7, line 10–12 in the revised manuscript (highlighted in yellow).

Figure R4. Reaction rate over the 15CuCe catalyst for six start-up cool-down cycles.

Comment 5: What about the reproducibility of this catalyst. Both from the point of view of the synthesis and the reaction.

Response: The reviewer’s valuable comment was appreciated. To ensure that the synthesis and catalytic performance of the 15CuCe catalyst were repeatable, another two batches of samples (sample 2 and sample 3) were synthesized by using the same method. As shown in Figure R5, the newly synthesized samples exhibited CO₂ conversion rates which were the same to that of the previously synthesized catalyst (sample 1), indicating the synthesis and catalytic performance of the 15CuCe catalyst were very repeatable.

Figure R5. Catalytic activities of the 15CuCe catalysts prepared in different batches.

Comment 6: Table 1 compares mostly Cu-based catalyst. Other transition metal catalyst should be compared in this table (for example Fe-based catalysts and more Ni-based catalyst).

Response: The reviewer's comment is highly appreciated. Based on the reviewer's suggestion, the catalytic performances of some other transition metal catalysts have been also compared with the CuCe catalysts in this work. According to the previous literatures, Fe-based catalysts and Ni-based catalysts have shown catalytic activity in the RWGS reaction, but their activity and CO selectivity still need to be improved. **As illustrated in Table R1, the CuCe catalysts in this work exhibited much higher activity and better CO selectivity than all the Fe-based and Ni-based catalysts which were reported in other literatures. The corresponding supplements have been added in the revised manuscript as Table 1 on page 6 (highlighted in yellow).**

Table R1. Comparison of CO₂ Conversion Rate and CO Selectivity for the as-prepared and Literature Reported catalysts.

Catalyst	H ₂ :CO ₂	Temperature (°C)	Pressure (MPa)	Rate (10 ⁻⁵ mol _{CO2} /g _{cat} /s)	CO selectivity (%)	Ref
15CuCe	3:1	600	0.1	146.6	100	this work
15CuAl	3:1	600	0.1	3.0	100	this work
Cu/CeO ₂ -hs	3:1	600	0.1	42.5	100	1
4Cu-Al ₂ O ₃	2:1	600	0.1	17.9	100	2
Cu/ β -Mo ₂ C	2:1	600	0.1	47.7	99.2	3
Cu-Fe/SiO ₂	1:1	600	0.1	11.9	100	4
NiCe/Zr	3:1	550	0.1	33.3	99.5	5
15CuCe	3:1	500	0.1	52.2	100	this work
In ₂ O ₃ -CeO ₂	1:1	500	0.1	2.98	100	6

K ₈₀ -Pt/L	1:1	500	0.1	2.22	100	7
Ni-in-Cu	3:1	500	0.1	3.95	100	8
CuSiO/CuO _x	3:1	500	0.1	3.18	100	9
TiO ₂ /Cu	3:1	500	0.1	1.78	N/A	9
SiO ₂ /Cu	3:1	500	0.1	1.11	N/A	9
Cu-Zn-Al	2:1	500	0.1	26.1	100	3
Cu/ β -Mo ₂ C	2:1	500	0.1	37.9	99.0	3
Ni/Mg(Al)O	3:1	450	0.1	0.5	66.7	10
Pt/TiO ₂	1:1	400	0.1	5.0	100	11
Pt/Al ₂ O ₃	1.4:1	400	N/A	0.16	N/A	12
Fe-CeO ₂	4:1	400	0.1	0.65	100	13

***Comment 7:** Regarding characterization, reduced samples' characterization should be mandatory in order to know exactly the morphology/chemistry of your sample before the reaction. It is more interesting to know the state of your active phases. Furthermore, during the text can be seen several affirmations about Cu being reduced during the reaction, but this statement is not correct since Cu was already reduced in the pre-activation treatment (described in the section "method")*

Response: In order to know exactly the structure of the 15CuCe catalyst after H₂ pretreatment, the HAADF-STEM and XPS measurements were conducted. As shown in Figure R6, ceria supports could maintain the rod-like morphology after pretreatment under H₂ flow, which indicated ceria nanorods have not undergone obvious sintering prior to the RWGS reaction. In addition, **2D and 3D copper clusters could be clearly observed on the surface of CeO₂ nanorods.** The width of the 2D layered clusters ranged from 1 nm to 3.5 nm, which was slightly smaller than the copper clusters after long-term RWGS reaction. And the average width and thickness of the 3D hemisphere shaped clusters were similar to that of the copper clusters after stability test, indicating the excellent stability of the copper clusters. **The Figure R6 has been shown in the revised supporting information as Figure S9 on**

page S14. The corresponding description has been added in the revised manuscript on *page 8, line 14–21* (highlighted in yellow).

Figure R6. (a, b, c, d) HAADF-STEM images of the pre-reduced 15CuCe catalyst.

As shown in Figure R7, the Cu 2*p* XPS spectra of the 15CuCe catalyst after H₂ pretreatment suggested the Cu²⁺ species was reduced to Cu⁺/Cu⁰ species. Thus, the statement about that Cu²⁺ was reduced to Cu⁺/Cu⁰ species during the RWGS reaction was not appropriated. **The Figure R7 has been added in the revised supporting information as Figure S16 on *page S21*, and the corresponding description has been shown in the revised manuscript on *page 11, line 2–9* (highlighted in yellow).** Thanks for the reviewer's comments again.

Figure R7. Cu 2p XPS results of the fresh, pre-reduced and used 15CuCe catalysts.

Comment 8: Homogeneity in the T units, no a mixture of °C and K (IR).

Response: We thank for the reviewer's comment. Sorry for missing the homogeneity in the T units. **The K has been changed to °C in the revised manuscript on page 12, line 6 and page 13, line 3 (highlighted in yellow).**

Comment 9: Synthesis. In the CeO₂ preparation procedure the solid obtained was not calcined until after the Cu impregnation, however, the alumina was calcinated previous to Cu deposition. This can bring differences in the support – Cu interaction.

Response: Thanks for the reviewer's valuable comments. In order to explore the effect of ceria calcination on the metal-support interaction, we prepared the CuCe catalyst with ceria support which was calcined at 600 °C for the Cu deposition. The obtained catalyst was denoted 15CuCe-600. As shown in Figure R8a, the TPR profile of the 15CuCe-600 catalyst could be deconvoluted into four peaks. The peaks denoted as α and β could be assigned to the progressive reduction of the dispersed CuO_x species to Cu⁺/Cu⁰ species. The γ peak was related to the Cu-[O_x]-Ce structures. And the θ peak could be attributed to the reduction of bulk CuO phase. Compared with the 15CuCe catalyst with ceria support which was not calcined previous to Cu deposition, the appearance of θ peak suggested that during the preparation of the catalyst, the change of the interaction between metal and support caused the increase of the size of

copper species. However, compared with the CeO₂ support and CuO, the significant shift of reduction peaks to low temperature meant that even though the CeO₂ support was calcined prior to loading copper, the interaction between copper and ceria in the prepared catalyst still maintained. Compared with the 15CuAl catalyst, the 15CuCe-600 catalyst still had much stronger metal-support interaction. Besides, the activity test result exhibited that the 15CuCe-600 catalyst had inferior activity than the 15CuCe catalyst (Figure R8b). Therefore, the catalyst with better catalytic activity could be prepared by depositing copper on uncalcined ceria nanorods. Thanks for the reviewer's valuable comments again.

Figure R8. (a) TPR profile and (b) CO₂ conversion rate over the 15CuCe-600 catalyst.

To Reviewer 2:

Comment 1: The work submitted by Prof. Jia, Prof. Ma and coworkers is an excellent study of rWGS reaction using an economical, active and stable Cu/CeO₂ catalyst. The results obtained are interesting and the mechanistic study proposed provides some relevant insights. However, there is not novelty and the impact is far from the objective of Nature Communications. Therefore, I recommend to submit this work to other journal specialized in catalysis such as J. Catal. or ACS Catal.

Response: Thanks for the reviewer's valuable comments and suggestions.

First of all, at present, the emission of CO₂ has become a serious environmental problem and the utilization of CO₂ has become a global focus. Thus, the development of the efficient RWGS reaction catalyst and the exploration of catalytic mechanism about CO₂ conversion have important research value in the fields of energy, environment and catalysis. Recently, many excellent works have systematically explored the structure-function relations of various catalysts. Liu et al. revealed the structure-function relation of the Ni-Au bimetallic nanoparticles in the RWGS reaction (W. Liu et al, *Nat. Catal.* **2020**, *3*, 411–417). J. Szanyi et al. explored the *in situ* dispersion process of Pd on TiO₂ during the RWGS reaction (J. Szanyi et al, *Angew. Chem. Int. Ed.* **2020**, *59*, 17657–17663). **However, the fabrication of the catalyst with both high activity and high stability under harsh reaction conditions is still urgently to be developed.**

Besides, as a kind of non-noble metal catalyst, copper–ceria catalyst showed catalytic activity in many catalytic reactions. **The exploration of structure and structure-function relation about copper–ceria catalyst is very important for the development of more efficient catalyst.** Recently, Shen et al. explored the size of copper species and interfacial structures in copper–ceria catalysts with low copper loading (~3 wt.%) after the reduction at different temperatures for only 2 hours (W. J. Shen et al, *Nat. Catal.* **2019**, *2*, 334–341). And our group also revealed the structure of the coordination-unsaturated and atomically dispersed copper species in the copper–ceria catalyst after air-calcination at 800 °C (C. J. Jia et al, *J. Am. Chem. Soc.* **2019**,

141, 17548–17557). However, **the structure of the copper–ceria catalyst, including the size of copper species and the morphology of ceria, is almost unknown under harsh reaction conditions (at high temperature and with reductive atmosphere), especially for the catalyst with relatively high copper loading.**

In addition, although many previous reports have pointed out that oxygen vacancy was related to the catalytic activity of CO₂ reduction, it is still unclear how the oxygen vacancy is involved in the reaction (J. Szanyi et al, *JACS Au*. **2021**, *1*, 977–986; C. W. Jones et al, *ACS Catal.* **2018**, *8*, 12056–12066).

For above existing scientific problems, we designed relevant experiments and carried out related researches. We think that it can meet the scope of your journal, and is justified for the *Nature Communications*. **We believe our findings will interest a broad readership in Catalysis and Materials Science according to the following aspects:**

1. We precisely determined the structure of the copper–ceria catalyst for the RWGS reaction under harsh reaction conditions.

The structure of the copper–ceria catalyst treated at high temperature (600 °C) and with reductive atmosphere was clearly revealed. **Under such a harsh reaction condition, the CeO₂ supports were only partially sintered, but not completely sintered.** With comprehensive characterizations, we found that **the interaction between copper and ceria maintained well after long-term catalytic tests**, which ensured **the copper species with high loading (~15 wt. %) were still very stable on the partially sintered support in the forms of 2D and 3D clusters.**

2. We directly explored the role of the oxygen vacancies in the RWGS reaction.

In situ Raman experiments were designed and carried out to investigate the role of oxygen vacancies. Based on the test results, we found that surface oxygen vacancies could be created by H₂ and consumed by CO₂. **The *in situ* generated and consumed surface oxygen vacancies in the copper–ceria catalyst were directly confirmed to be involved in the RWGS reaction.**

3. We clearly revealed the synergistic effect between copper cluster and oxygen vacancy.

Based on the experimental results and DFT calculations, we found that **the surface oxygen vacancy induced synergistic effect with the adjacent copper cluster, improving the activation of CO₂ and the formation of active intermediates. The copper cluster-oxygen vacancy synergistic sites ensured the copper–ceria catalyst had unmatched activity to catalyze the RWGS reaction, which surpassing almost all other reported metal catalysts.** Besides, this copper–ceria catalyst also maintained excellent stability in the 240 h activity evaluation under very harsh reaction conditions (600 °C, space velocity of 400,000 mL·g_{cat}⁻¹·h⁻¹).

After all, in this work, the efficient copper–ceria with both high activity and stability under very harsh reaction conditions was constructed. And the structure-function relation of the partially sintered copper–ceria catalyst was clearly revealed. **We think these findings can provide a reference for the exploration and application of sintered catalyst in the high temperature reaction.** We are looking forward to your next comments.

To Reviewer 3:

Reviewer #3: Key results:

In this research study, Liu et al. have designed a Cu-based catalyst with very high activity and stability. The catalyst performs better than previously reported Cu-based catalysts for the RWGS reaction. They also showed the effect of various Cu loading, the role of oxygen vacancies in the reaction, and the SMSI effect on the catalyst activity.

Response: We thank for the reviewer's comments. **In this work, the partially sintered copper–ceria catalyst was constructed, which exhibited excellent catalytic activity and high stability in the RWGS reaction, surpassing almost all other reported non-noble metal catalysts and even costly noble metal catalysts.** Even though the CeO₂ support suffered severe sintering, the interaction between copper and ceria could maintained well after the long time treatment under high temperature and reductive atmosphere, which ensured the copper clusters stabilized on the partially sintered CeO₂ nanorods.

Validity:

The data and results are valid and the data seem to be collected and interpreted in an acceptable way based on the description of the experiments.

Response: Thanks for the reviewer's valuable comments.

Significance:

The findings are important due to the high activity and stability of the proposed catalyst for the reverse water gas shift reaction. However, Cu-ceria is an obvious system to prepare and test for scientists in this community. We have prepared copper on standard ceria support (no nanorods!) by wet impregnation and the activity was clearly below that of Cu-alumina. The main question why this specific catalyst is so much more active than other catalysts, particularly other Cu-ceria catalysts, has not been addressed by the authors, which limits the value of the paper for the scientific

community. It is not clear what the reader should learn about the preparation of RWGS catalysts, besides that this very special catalyst is very active. The described material characterization data is solid and clear but does not reveal any principally new insight on this catalyst type, which one would expect from a publication in Nature Communications. Therefore, I strongly recommend the authors to conduct additional experiments to elaborate the difference between their new catalyst and other RWGS catalysts, particularly standard Cu-ceria catalysts with the same composition.

Response: The reviewer's valuable comment was highly appreciated by us. The significance of this work is as follows:

1. In this work, the partially sintered copper–ceria catalyst with sufficient and stable copper sites was constructed, which exhibited excellent catalytic performance under the very harsh conditions. **The abundant and stable copper clusters with sufficient surface vacancies accounted for the extraordinary activity and stability. We found that the interaction between copper and ceria maintained well after long-term RWGS reaction, which ensured the copper species with high loading (~15 wt. %) stabilized on the partially sintered CeO₂ nanorods in the forms of 2D and 3D clusters.** According to the reviewer's suggestions, standard copper–ceria catalysts with the same composition were synthesized by using ceria nanocube and ceria nanoparticle as supports. Based on the characterization results and activity evaluation, the copper–ceria catalyst synthesized by ceria nanorod showed unique structural advantage and better catalytic performance (Figure R9). **On CeO₂ nanorods, copper species were more dispersed and oxygen vacancies were easier to form, which made the copper–ceria catalyst with CeO₂ nanorods as support have the best initial activity.** Besides, the CeO₂ nanorods had better structural stability (Table R2), which prevented the inactivation caused by the excessive sintering of the catalyst. In addition, the standard copper–ceria catalyst prepared by conventional impregnation (IMP) method had much inferior activity due to the poor dispersion of copper species (Figure R11). **Copper species were easier to be**

anchored on the ceria support by DP method rather than IMP method (Figure R11).

2. The active sites in supported catalysts are apt to sinter on-stream, especially for the high-temperature reactions. In this work, we utilized the interaction between active metal and support to construct abundant active copper clusters on the partially sintered ceria nanorod support, which exhibited high activity and stability in the high-temperature RWGS reaction. The construction of partially sintered catalyst provided a valuable strategy for the development of efficient catalysts in high-temperature reactions. In addition, the exploration of the structure and the structure-function relation about copper–ceria catalyst is very important for the development of more efficient catalysts. Recently, Shen et al. explored the size of copper species and interfacial structures in copper–ceria catalysts with low copper loading (~3 wt.%) after the reduction at different temperatures for only 2 hours (W. J. Shen et al, *Nat. Catal.* **2019**, *2*, 334–341). And our group also revealed the structure of the coordination-unsaturated and atomically dispersed copper species in the copper–ceria catalyst after air-calcination at 800 °C (C. J. Jia et al, *J. Am. Chem. Soc.* **2019**, *141*, 17548–17557). **However, the structure of the copper–ceria catalyst, including the size of copper species and the morphology of ceria, is almost unknown under harsh reaction conditions (at high temperature and with reductive atmosphere), especially for the catalyst with relatively high copper loading. In this work, we precisely determined the structure of the copper–ceria catalyst for the RWGS reaction under harsh reaction conditions, which also provided a reference for the development of the copper–ceria catalyst.**

Data and Methodology:

While the findings are interesting, I think some of the arguments are not at the level expected for publication at Nature Communication.

Response: Thanks for the reviewer’s comments. The reviewer’s comments and suggestions are very important for the improvement of our research work. According

to the review's suggestions, we supplemented the corresponding experiments and DFT calculations, further deepen the analysis of experimental data. We are looking forward to your next comments.

***Comment 1:** The authors indicated that they used CeO₂ nanorods but these nanorods deformed/transformed through the reaction. Why using nanorods are important if the morphology changes during the reaction? What happens if we use other morphologies of CeO₂? How does the new morphology affect the reaction? Do we know the surface area of the used catalyst? Does the surface area increase or decrease after the change of morphology? Why does the size of the CeO₂ crystals, counterintuitively, reduce after 70 h of reaction (based on the XRD patterns in Figure S7c)?*

Response: The reviewer's comment is highly appreciated by us. In order to respond to the reviewer's comment, we have done additional experiments, analyzed and summarized the results as below:

1. The influence of CeO₂ morphology:

(1) In order to explore the morphological effects of the ceria supports on catalytic performance, the CeO₂ nanocubes and nanoparticles were synthesized and used as supports. The obtained catalysts were donated as 15CuCe-NC and 15CuCe-NP, where the 15 was the copper loading in weight percent (15 wt.%). The corresponding preparation details were added in the revised supporting information on *page S3, line 2–11* (highlighted in yellow). As illustrated in Figure R9a, **the CO₂ conversion rate ranked in the order of 15CuCe-NR > 15CuCe-NP > 15CuCe-NC.** And all three catalysts showed 100% CO selectivity. It has been reported that CeO₂ nanorod had lower oxygen vacancy formation energy than CeO₂ nanocube and CeO₂ nanoparticle (X. H. Liu et al, *ACS Catal.* **2020**, *10*, 11493–11509; G. Z. Lu et al, *ACS Catal.* **2016**, *6*, 2265–2279). From the Raman results of the fresh catalysts, the peaks assigned as oxygen vacancy in the 15CuCe-NR catalyst were stronger than that in the 15CuCe-NC and 15CuCe-NP catalysts. Higher oxygen vacancy concentration could improve the initial activity of the 15CuCe-NR catalyst. And as shown in the XRD results of the fresh copper–ceria catalysts with different morphological ceria supports

(Figure R9d), the 15CuCe-NC catalyst showed more distinct diffraction peaks of CuO than 15CuCe-NR and 15CuCe-NP catalysts, which indicated the copper species had worse dispersion on ceria nanocube, causing the much inferior catalytic performance. The TEM pictures also reflected that there were many copper agglomerations (labeled by circle in red) on the ceria nanocube, but not on ceria nanoparticle, which was consistent with the XRD results. **Highly dispersed copper species and higher concentration of oxygen vacancies made the 15CuCe-NR had the best initial RWGS reaction performance. The Figure R9 has been added in the revised supporting information as Figure S4 on page S9, and the corresponding description has been shown in the revised manuscript on page 6, line 10–13 (highlighted in yellow).** Thanks for the reviewer again.

Figure R9. (a) CO₂ conversion and (b) CO selectivity of the 15CuCe-NR, 15CuCe-NP and 15CuCe-NC catalysts. (c) Raman spectra and XRD results over the 15CuCe-NR, 15CuCe-NP and 15CuCe-NC catalysts. (e,f) TEM pictures of the fresh 15CuCe-NP and 15CuCe-NC catalysts, respectively.

(2) The specific BET surface areas (Table R2) suggested that the used 15CuCe-NR catalyst had a much larger specific surface area than that of the used 15CuCe-NP catalyst, which confirmed that the CeO₂ nanorod had better structural stability than the CeO₂ nanoparticle. In our previous report, it was

proved that the CeO₂ nanorod was more resistant to sintering than the CeO₂ nanoparticle during the air-calcination at high temperature (C. J. Jia et al, *J. Am. Chem. Soc.* **2019**, *141*, 17548–17557). Excessive sintering of the catalyst will lead to the deactivation. As shown in Figure R10, the stability evaluation of the 15CuCe-NP catalyst indicated that it lost 30% of its origin activity within 20 h. **The Figure R10 has been shown in the revised supporting information as Figure S5 on page S10, and the responding description has been added in the revised manuscript on page 7, line 7–9 (highlighted in yellow).** To sum up, ceria nanorod was a better choice to prepare copper–ceria catalyst with more efficient catalytic activity in the high-temperature RWGS reaction, when compared with the ceria with other morphologies (nanoparticle and nanocube). Thanks for the reviewer again.

Table R2. BET specific surface areas of various copper–ceria catalysts.

Catalyst	S _{BET} (m ² g _{cat} ⁻¹)
15CuCe-NR	^α 83.3, ^β 44.1
15CuCe-NP	^α 82.3, ^γ 27.9
15CuCe-NC	^α 23.3
15CuCe-IMP	^α 60.3

Note: (α) fresh catalysts. (β, γ) used catalysts after 70 h and 40 h RWGS reaction, respectively.

Figure R10. Stability tests of the 15CuCe-NP and 15CuCe-NR catalysts.

2. The influence of preparation method:

To explore the importance of the preparation method, we synthesized the reference catalyst by typical impregnation (IMP) method. The reference catalyst was denoted 15CuCe-IMP. As shown in Figure R11a, the 15CuCe-IMP exhibited much lower RWGS conversion than that the 15CuCe prepared by DP method. The distinct diffraction peaks of CuO in the XRD result (Figure R11b) indicated the worse dispersion of copper species for the fresh 15CuCe-IMP sample. And the TEM picture of the fresh 15CuCe-IMP sample (Figure R11c) illustrated that the ceria nanorod suffered more severe sintering during the preparation of catalyst, causing the decrease of the specific surface area ($60.3 \text{ m}^2\cdot\text{g}^{-1}$) (Table R2). **The interaction between copper and ceria was relatively weak, which could make neither the active metal, nor the ceria support stable.** The copper species with poor dispersion also accelerated the sintering of ceria. Above data demonstrated that for the copper–ceria catalyst, the dispersion degree of copper species had an important effect on the activity and structural stability. From the perspective of catalyst synthesis method, copper was easier to be anchored on the ceria support by DP method rather than IMP method. **The data has been added in the revised supporting information as Figure S14 on page S19, and the corresponding description has been shown in the revised manuscript on page 10, line 14–16 (highlighted in yellow).** Thanks for the reviewer again.

Figure R11. (a) CO₂ conversion of the 15CuCe-IMP catalyst. (b) XRD and (c) TEM results of the 15CuCe-IMP catalyst, respectively.

3. The size of the CeO₂ crystals:

The reviewer mentioned that the size of CeO₂ crystal reduced after reaction for 70 h. Actually, the size of the CeO₂ crystals increased after stability test. The weak diffraction peaks might be caused by using too little sample (<10 mg) in the XRD measurement of the used 15CuCe catalyst. In order to get better data, more sample after 70 h of RWGS reaction was used to repeat the XRD test. As shown in figure R12, compared with the fresh 15CuCe catalyst, the CeO₂ diffraction peaks of the used sample became sharper, and the half-maximum width became narrower, which suggested the size of CeO₂ crystal increased. **Furthermore, determined by the XRD patterns and Scherrer formula, the mean crystallite size of CeO₂ grew from 10.2 nm to 17.2 nm after long-term RWGS reaction. The data has been added in the revised supporting information as Figure S13c on page S18 (highlighted in yellow).** Thanks for the reviewer again.

Figure R12. The XRD patterns of the fresh and used catalysts.

Comment 2: CO Selectivity is defined as the CO concentration over the total concentration of CH₄ and CO leaving the reactor. This either should be changed to $nCO/(nCO_{2in} - nCO_{2out})$ to account for all possible products of the reaction (which might not have been detected), or there must be a carbon balance accompanied by TPO to make sure the system's carbon balance is valid.

Response: We thank for the reviewer's comment. In order to detect other products that might be existed in addition to CH₄ and CO, the products and reactants in the RWGS reaction catalyzed by the 15CuCe catalyst were further detected online by using two tandem gas chromatography equipped with a flame ionization detector (FID) and a thermal conductivity detector (TCD). In the catalytic performance evaluation, no other products were detected besides CO, which again confirmed the CO selectivity was 100%. Besides, the carbon balance was found to be greater than 97% in all the tests. Carbon balance below 100% might be due to the carbon deposition. From the Raman results in Figure R13a, there were no obvious peaks of carbon deposition over the used 15CuCe catalyst after 20 h RWGS reaction at 600 °C. And as shown in Figure R13b, CO₂ desorption peaks appeared in the O₂-TPO and Ar-TPD results, which suggesting the CO₂ signal in the O₂-TPO result might come from the CO₂ desorption rather than carbon deposition. Thus, the carbon deposition on the catalyst surface was not severe during the RWGS reaction. Thanks for the reviewer again.

Figure R13. (a) Raman spectra and (b) Ar-TPD and O₂-TPO profiles over the used 15CuCe catalyst after 20 h RWGS reaction, respectively.

Comment 3: Through Raman spectroscopy, it is shown that CO₂ is adsorbed on the oxygen vacancies. But no data on how CO₂ is adsorbed is presented. In-situ DRIFTS for CO₂ adsorption is needed so that we can see how CO₂ is adsorbed. The fact that

CO₂ is adsorbed on the oxygen vacancies does not rule out the other possible active sites for CO₂ adsorption.

Response: Thanks for the reviewer's valuable comments. The *in situ* DRIFTS spectra for CO₂ adsorption under CO₂ treatment and RWGS reaction conditions were given in Figure R14. In Figure R14a, the bands located at around 1588 and 1286 cm⁻¹ were usually related to bidentate carbonate (C. W. Jones et al., *ACS Catal.* **2018**, *8*, 12056–12066; D. Ma et al., *J. Phys. Chem. C* **2018**, *122*, 12934–12943), which suggested that CO₂ was adsorbed on the surface of the catalyst to form carbonate species. In order to monitor the generation of carbon species further, the *in situ* DRIFTS was also measured for the 15CuCe catalyst under the RWGS reaction. As illustrated in Figure R14b, in addition to the signals of bidentate carbonate (1286, 1583 cm⁻¹) and polydentate carbonate (1330 cm⁻¹), the band of formate (1373 cm⁻¹) was also observed (D. Q. Ye et al., *Appl. Surf. Sci.* **2020**, *516*, 146035). The *in situ* DRIFTS result could give relevant information of adsorbed species, but could not reflect the specific active sites for CO₂ adsorption. **The corresponding supplements have been added in the revised supporting information as Figure S26 on page S31, and the corresponding description has been shown in the revised manuscript on page 17, line 8–14 (highlighted in yellow).**

To further verify the adsorption sites for CO₂, calculations based on density functional theory (DFT) were performed. The adsorption characteristics of CO₂ indicated that the presence of Cu atoms assured that the decreasing entropy step could occur, shown as Figure R15 and Table R3. Notwithstanding the situation of oxygen vacancy affected the binding force, **the energy could be reduced more than one electron-volt under the bonding interaction between Cu and CO₂, suggesting the oxygen vacancy formed the synergistic catalytic effect with the adjacent copper cluster to promote the adsorption of CO₂.** Therefore, compared with the sole oxygen vacancy site, the CO₂ molecule preferentially adsorbs on the interface between copper cluster and oxygen vacancy. Thanks for the reviewer again.

Figure R14. *In situ* DRIFTS spectra of the 15CuCe catalyst during (a) CO₂ treatment and (b) RWGS reaction conditions at 300 °C.

Figure R15. (a) Chemisorption of CO₂ on the 10Cu/CeO₂{111} surface. Five oxygen vacancies, named after V_O-A to V_O-E, were made comparisons, and the selected CO₂ was located close to V_O-A. (b) The adsorption energy that CO₂ was bound to V_O on the CeO₂{111} surface was obviously weaker than those of Cu dropped ceria sites.

Table R3. The single-point energies of the constructed surface models and intermediates.

Structures	E / eV	Structures	E / eV
V _O -A	-1218.133569	V _O -A + CO ₂	-1243.052508
V _O -B	-1217.825478	V _O -B + CO ₂	-1241.988960
V _O -C	-1217.889766	V _O -C + CO ₂	-1242.117825

V _O -D	-1217.569036	V _O -D + CO ₂	-1242.113886
V _O -E	-1218.779069	V _O -E + CO ₂	-1243.594793
V _O /CeO ₂ {111}	-1184.992895	V _O /CeO ₂ {111} + CO ₂	-1207.824954
IMA1	-1243.052508	IMA2	-1250.479925
IMA3	-1250.915264	IMA4	-1250.805777
IMA5	-1249.218921	IMA6	-1234.602513
IMA3-I	-1250.773474	IMA4-I	-1250.782227
IMA3-II	-1250.717540	IMA4-II	-1250.340765
CO ₂	-23.014969	H ₂	-6.762122
H ₂ O	-14.232397		

Analytical Approach:

Comment 1: *The reaction mechanism study is weak. It clearly shows that the redox mechanism is not happening. But the sole detection of formate on the surface is not enough to conclude about the reaction pathway. There can be various CO formation routes such as the carbonate route and hydroxycarbonyl route. Formate can also be a spectator (or minor reaction intermediates) on the surface. The full spectra should be visible to see the formed species. Better methodologies (such as addition of DFT study, Modulation Excitation Spectroscopy, or other in-situ methodologies) should be used to prove the reaction mechanism.*

Response: The reviewer's comment is highly appreciated by us. As shown in Figure R14a, when the sample was exposed to the CO₂ gas flow, even with the formation of carbonate species, no gaseous signal of CO was detected, which meant carbonate species might not be converted directly to CO. **According to the reviewer's suggestion, the relevant DFT study was performed to explore the reaction**

mechanism further. As shown in Figure R15, CO₂ was more easily adsorbed at V_O-A. In the presence of H₂, Cu atoms captured H₂ molecule and broken H-H bond, then transferred H atom to CO₂, shown as Figure R16. Formate structures formed accompanying the formation of C-H bond, and these structures exhibited in the intermediate IMA3, IMA3-I, IMA4 and IMA4-I. The heat liberation declared that the formation of the formate was a thermodynamic feasible elementary reaction. The subsequent hydrogen-migration step ($\Delta E = 1.587$ eV) was the thermodynamically limiting step, and this might be the reason that formate signals were detected by DRIFTS. Carboxylic intermediates, IMA3-II and IMA4-II, were involved in the mechanism at the same time. Different from the formate intermediates, one step was absent from the carboxylic path, i.e., IMA4-II produced IMA6 directly. **The Figure R15 and R16 have been added in the revised manuscript as Figure 7 on page 18, and the corresponding description has been added in the revised manuscript on page 17, line 17–30 and page 18, line 1–7 (highlighted in yellow). The Table R3 has been added in the revised supporting information as Table S2 on page S33 (highlighted in yellow).** Thanks for the reviewer again.

Figure R16. RWGS reaction mechanism occurred in the V_O-A. The red, blue and black lines indicated different reaction paths, and the structural diagrams with the red, blue or black stroke corresponded to the reaction paths, respectively.

Comment 2: *There are more than two peaks in the H₂-TPR profiles of the catalysts. These peaks need to be identified and described.*

Response: Thanks for the reviewer's comment. In the previous report, the EXAFS data confirmed the existence of Cu-O and Cu-Ce binding in the Cu/CeO₂-NR catalyst, which was quite consistent with the reduction peaks of CuO_x clusters and the Cu-[O_x]-Ce structure (W. W. Wang et al, *ACS Catal.* **2017**, 7, 1313–1329). However, the experimental result in the Figure R17 and previous reports indicated that the H₂-TPR pattern of pure CuO was not completely symmetric (H. Yan et al, *Appl. Catal. B Environ.* **2018**, 226, 182–193), which indicated that the reduction of CuO_x was not completed in one step, suggesting the CuO_x species were progressively reduced to Cu⁺ and Cu⁰ species. Besides, the reduction peak of CuO could not be deconvoluted into two similar peaks, indicated the reduction of Cu²⁺ to Cu⁺ and the reduction of Cu⁺ to Cu⁰ occurred simultaneously in a certain range of reduction temperature. Therefore, we speculated the reduction peaks of highly dispersed CuO_x clusters in Cu/CeO₂-NR catalysts were also not symmetric, and the α and β peaks could not be attributed to the reduction of single species, but the progressively reduction of CuO_x species to Cu⁺ and Cu⁰ species (Figure R18). Furthermore, the high-temperature γ peak (170–260 °C) was due to the reduction of the strong interacted Cu-[O_x]-Ce structure. **The corresponding supplement has been added in the revised supporting information as Figure S17 on page S22, and the corresponding description has been added in the revised manuscript on page 11, line 14–26 (highlighted in yellow).** Thanks for the reviewer again.

Figure R17. The H₂-TPR profile of pure CuO.

Figure R18. H₂-TPR profiles of all copper–ceria catalysts.

Comment 3: Why do we observe a weaker signal for XPS after reaction (Figure S9)? Does Cu diffuse into CeO₂? What is the state of the signal for the H₂-treated catalyst before the reaction?

Response: Thanks for the reviewer's comment. We would like to respond in two aspects:

1. The weaker signal of the XPS after reaction was caused by the poor signal noise ratio of the data. According to previous reports, if Cu²⁺ ions were incorporated into ceria lattice to form solid solution, the position of CeO₂ diffraction peaks could be shifted to high 2θ values due to the radius of Cu²⁺ ions (0.072 nm) was smaller than that of Ce⁴⁺ ions (0.101 nm) (C. Li et al. *Chem. Mater.* **2003**, *15*, 4761–4767). From the XRD results of the fresh and used 15CuCe catalysts in Figure R19, the XRD peaks of CeO₂ did not shift to higher angle, indicating the Cu species were supported on CeO₂ support rather than incorporated into the CeO₂ lattice. In addition, the HAADF images also suggested copper species were supported on the surface of ceria.

Figure R19. XRD patterns of the CeO₂ supports and the 15CuCe catalysts.

2. The Cu 2*p* XPS spectra of the 15CuCe catalyst after H₂ pretreatment was shown in Figure R20. Compared to the fresh catalyst, **the 15CuCe catalyst exhibited a significant reduction with Cu²⁺ transforming into Cu⁺/Cu⁰ during the H₂ pretreatment.** Thanks for the reviewer again.

Figure R20. Cu 2*p* XPS results of (a) fresh, (b) H₂-pretreated and (c) used 15CuCe catalysts.

Comment 4: *It is not clear why there are less oxygen vacancies created when testing at 500 °C as opposed to 300 °C (Figure 4a).*

Response: Thanks for the reviewer's comment. As illustrated in the H₂-TPR profile of the 15CuCe catalyst (Figure R21), **there was no any reduction peak between 300 °C to 500 °C, which indicated that under the reducing atmosphere, no more oxygen atoms in the CeO₂ support could be reduced to form oxygen vacancies even if the test temperature was increased from 300 °C to 500 °C.** In the *in situ* Raman tests under the reaction conditions of 300 °C and 500 °C, the small difference

in the concentration of the oxygen vacancy could be considered within the error range, which could not confirm less oxygen vacancies were created when testing at 500 °C as opposed to 300 °C. **The corresponding description has been added in the revised manuscript on page 14, line 23 and page 15, line 1–3 (highlighted in yellow).** Thanks for the reviewer’s valuable comment again.

Figure R21. H₂-TPR profile of the 15CuCe catalyst.

***Comment 5:** In the last paragraph on page 14, the authors indicate that the lower apparent reaction order of CO₂ and H₂ on 15CuCe compared to 5CuCe shows the higher ability of this catalyst in adsorption and activation of CO₂ and H₂. This conclusion is not clear to me. What we can understand from the lower activation of H₂ and CO₂ on 15CuCe is that the reaction rate on this catalyst is less dependent on the concentration of CO₂ and H₂. This does not lead to the conclusion that CO₂ and H₂ can better be adsorbed and activated for the RWGS reaction. Stronger adsorption of CO₂ or H₂ does not necessarily lead to their higher activity or higher level of participation in the reaction. The authors need to clarify their statements and/or add supporting arguments.*

Response: Thanks for the reviewer’s comment. The previous statement in the manuscript about the apparent reaction orders of CO₂ and H₂ was inappropriate. The reaction orders of CO₂ for the 15CuCe and 5CuCe catalysts were 0.25 and 0.52, respectively. And the reaction orders of H₂ over the 15CuCe and 5CuCe catalysts

were 0.25 and 0.3, respectively. **The lower apparent reaction orders of CO₂ and H₂ on the 15CuCe catalyst compared to the 5CuCe sample reflected that the reaction rate on the 15CuCe catalyst was less dependent on the concentrations of CO₂ and H₂, which might suggested CO₂ and H₂ were relatively easily adsorbed on the 15CuCe catalyst with more oxygen vacancies and copper sites.** The reaction orders do not reflect the ability of the catalyst in the activation of reactant molecules. **The previous statement has been revised, and the corresponding description has been added in the revised manuscript on page 16, line 1–7 (highlighted in yellow).** Thanks for the reviewer again.

Suggested Improvements:

Most suggested improvements are already listed above. However, some additional suggestions are shown below:

Comment 1: *Another test with a catalyst with Cu loading between 15 and 25 might be needed to see if 15% Cu is the optimum loading.*

Response: In order to explore whether 15% was the optimal loading, the activity tests of the 18CuCe and 22CuCe catalysts were measured. As shown in Figure R22, the 18CuCe and 22CuCe catalysts showed lower activity than the 15CuCe catalyst. Therefore, based on existing experimental results, 15% Cu was the optimum loading. Thanks for the reviewer again.

Figure R22. Activities of the 15CuCe, 18CuCe and 22CuCe catalysts.

Comment 2: *In Figure 4a and Figure S14, the insets should be explained.*

Response: Thanks for the reviewer's reminding. **The corresponding description has been added in the revised manuscript on page 14, line 23 and page 15, line 1–3 (highlighted in yellow).**

Comment 3: The text need full revision. There are grammatical errors (“strong interacted... structure”, “the reductive support with oxygen vacancies was easier to activate CO₂”, first three lines of page 12, “the DI band was even obvious than F₂g peak during...”, “have no influence to the intrinsic defects”, etc.) as well as typos (Figure 3 caption, etc.) which need to be fixed.

Response: Thanks for the reviewer's comment. The corresponding description has been changed in the revised manuscript (highlighted in yellow).

References

1. Zhang, Y. et al. Highly efficient Cu/CeO₂-hollow nanospheres catalyst for the reverse water-gas shift reaction: Investigation on the role of oxygen vacancies through in situ UV-Raman and DRIFTS. *Appl. Surf. Sci.* **516**, 146035 (2020).
2. Bahmanpour, A. M. et al. Cu-Al Spinel as a Highly Active and Stable Catalyst for the Reverse Water Gas Shift Reaction. *ACS Catal.* **9**, 6243–6251 (2019).
3. Zhang, X. et al. Highly Dispersed Copper over β -Mo₂C as an Efficient and Stable Catalyst for the Reverse Water Gas Shift (RWGS) Reaction. *ACS Catal.* **7**, 912–918 (2016).
4. Chen, C. Study of iron-promoted Cu/SiO₂ catalyst on high temperature reverse water gas shift reaction. *Applied Catalysis A: General.* **257**, 97–106 (2004).
5. Zonetti, P. C. et al. The Ni_xCe_{0.75}Zr_{0.25-x}O₂ solid solution and the RWGS. *Applied Catalysis A: General.* **475**, 48–54 (2014).
6. Wang, W. et al. Reverse water gas shift over In₂O₃-CeO₂ catalysts. *Catal. Today.* **259**, 402–408 (2016).
7. Yang, X. et al. Promotion effects of potassium on the activity and selectivity of Pt/zeolite catalysts for reverse water gas shift reaction. *Applied Catalysis B: Environmental.* **216**, 95–105 (2017).
8. Wang, L. et al. Dispersed Nickel Boosts Catalysis by Copper in CO₂ Hydrogenation. *ACS Catal.* **10**, 9261–9270 (2020).
9. Yu, Y. et al. Highly active and stable copper catalysts derived from copper silicate double-shell nanofibers with strong metal–support interactions for the RWGS reaction. *Chem. Commun.* **55**, 4178–4181 (2019).
10. Rodrigues, M. T. et al. RWGS reaction employing Ni/Mg(Al,Ni)O–The role of the O vacancies. *Applied Catalysis A: General.* **543**, 98–103 (2017).
11. Chen, X. et al. Catalytic performance of the Pt/TiO₂ catalysts in reverse water gas shift reaction: Controlled product selectivity and a mechanism study. *Catal. Today.* **281**, 312–318 (2017).

12. Kim, S. S., Lee, H. H. & Hong, S. C. A study on the effect of support's reducibility on the reverse water-gas shift reaction over Pt catalysts. *Applied Catalysis A: General*. **423–424**, 100–107 (2012).
13. Xiong, K. et al. CO₂ Reverse Water-Gas Shift Reaction on Mesoporous M-CeO₂ Catalysts. *The Canadian Journal of Chemical Engineering*. **95**, 634–642 (2017).

REVIEWER COMMENTS

Reviewer #1 (Remarks to the Author):

Authors have made a significant effort to address most of the points indicated by the referees. Hence the paper has been remarkably improved.

From this reviewer's perspective, I am satisfied with the responses for all my comments except for the first one related to equilibrium data. Authors are deliberately ignoring the competing process: CO₂ methanation on the basis of the observed full selectivity to CO in the experiments. This is not convincing and it is in fact a wrong practice to assume that experimental data could give fundamental process equilibrium trends. Experimental conversion and equilibrium conversion must be decoupled and treated as separate part of the story. The fact that experimentally authors do not observe methane does not change at all the underlying equilibrium thermodynamics and hence the represented equilibrium plot is not correct. There are many references in literature including the correct equilibrium plot.

Despite the manuscript improvement, I stand by my initial assessment and I believe this paper is more suitable for a specialized journal in catalysis and not suitable for Nature Comm. It does not level up the standards of the journal and it is not novel enough and does not have sufficient impact for Nature Comm. Actually, the paper provides incremental advances rather than innovative results.

Reviewer #3 (Remarks to the Author):

The authors have carefully revised the manuscript according to the reviewer's comments. They have seized all suggestions and significantly improved the manuscript.

Although the study is very comprehensive and explains the behavior and the structure of the catalyst with great detail, the studied catalyst suffers from somewhat limited technical relevance, since ceria nanorods are unlikely to be applied as support material in industrial processes. Nevertheless, the results are of sufficient novelty and detail to justify publication in Nature Communications.

I have the following comments that should be considered by the authors:

In Figure R1, it is unclear to which axis the shown bars refer.

In Figure R3 and R6, the differentiation between 2D and 3D clusters is not convincing. Due to the cylindrical shape of the nanorods, 3D clusters distributed on the surface of the nanorods could appear as flat or two-dimensional (2D) depending on the viewing angle. Please provide more convincing results and arguments for the claimed 2D clusters.

Responses to the Reviewers' Comments and the Corresponding Revisions

To Reviewer #1

Comment: Authors have made a significant effort to address most of the points indicated by the referees. Hence the paper has been remarkably improved.

From this reviewer's perspective, I am satisfied with the responses for all my comments except for the first one related to equilibrium data. Authors are deliberately ignoring the competing process: CO₂ methanation on the basis of the observed full selectivity to CO in the experiments. This is not convincing and it is in fact a wrong practice to assume that experimental data could give fundamental process equilibrium trends. Experimental conversion and equilibrium conversion must be decoupled and treated as separate part of the story. The fact that experimentally authors do not observe methane does not change at all the underlying equilibrium thermodynamics and hence the represented equilibrium plot is not correct. There are many references in literature including the correct equilibrium plot.

Despite the manuscript improvement, I stand by my initial assessment and I believe this paper is more suitable for a specialized journal in catalysis and not suitable for Nature Comm. It does not level up the standards of the journal and it is not novel enough and does not have sufficient impact for Nature Comm. Actually, the paper provides incremental advances rather than innovative results.

Response: According to the previous comments and suggestions of the reviewer, we have supplemented relevant experiments and carefully revised the manuscript, which undoubtedly brought great help for the improvement of our work. Therefore, thanks for the reviewer's valuable comments and suggestions on our work. Actually, **the partially sintered copper–ceria catalyst developed in this work improved the high-temperature RWGS reaction performance to a new level, which showed great potentials in the catalysis applications. We believe that our work is not only limited to catalysis, but will also inspire researchers working in general chemistry, material sciences, and energy-related fields. Therefore, we think it is**

suitable for publication on *Nature Communications*. We would like reply to the reviewer in detail as described as below:

Comment 1: Authors are deliberately ignoring the competing process: CO₂ methanation on the basis of the observed full selectivity to CO in the experiments. This is not convincing and it is in fact a wrong practice to assume that experimental data could give fundamental process equilibrium trends. Experimental conversion and equilibrium conversion must be decoupled and treated as separate part of the story. The fact that experimentally authors do not observe methane does not change at all the underlying equilibrium thermodynamics and hence the represented equilibrium plot is not correct. There are many references in literature including the correct equilibrium plot.

Response: Thanks for the reviewer's comment. In the calculation of equilibrium limitation, we only considered the RWGS reaction, which was incomplete. Although methane was not observed in the products, the underlying equilibrium thermodynamics should not be ignored. **The corrected CO₂ thermodynamic equilibrium conversion was calculated from HSC chemistry software version 6.0. The ratio of H₂:CO₂ in the initial state was 3:1, and the products including CO, H₂O and CH₄ were taken into account in the calculation process.** The methanation and RWGS reaction have been considered in the simulation process. As shown in Figure R1, the thermodynamic equilibrium curve indicated the natural competition between methanation and RWGS reaction. In the low-temperature range, high CO₂ conversion can be achieved by methanation. And in the high-temperature range, the RWGS reaction becomes the predominant reaction^{1,2}. Compared to the CO₂ equilibrium conversion in previous manuscript (Figure R1a), the corrected equilibrium curve shows a much higher CO₂ conversion in the low temperature range due to the methanation (Figure 1b). **The Figure R1 has been added as Figure 1a in the revised manuscript on page 5. And the corresponding description has been shown in page 5, line 10–13 and page 20, line 22–24 in the revised manuscript (highlighted in yellow).** Thanks for the reviewer's comment again.

Figure R1. (a) Equilibrium limitation and CO₂ conversion of CeO₂, 5CuCe, 15CuCe and 15CuAl catalysts in previous manuscript; (b) Corrected equilibrium limitation and CO₂ conversion of CeO₂, 5CuCe, 15CuCe and 15CuAl catalysts.

Comment 2: Despite the manuscript improvement, I stand by my initial assessment and I believe this paper is more suitable for a specialized journal in catalysis and not suitable for Nature Comm. It does not level up the standards of the journal and it is not novel enough and does not have sufficient impact for Nature Comm. Actually, the paper provides incremental advances rather than innovative results.

Response: As we have replied to the reviewer last time, we think our work not only give important findings in catalysis, but also will inspire researchers working in synthetic chemistry and material sciences, and energy-related fields. Since the reviewer still have doubts on the innovation on our work. We would like to emphasize the significance and innovation on our work in detail again. **We do think our findings and innovations provided solutions to the following crucial scientific questions:**

1. Can stable and efficient supported copper-based catalyst be constructed under very high temperature and with reductive atmosphere and what about the precise structure of the catalyst?

Supported copper-based catalyst as a typical representative has been used in a variety of important reactions, such as CO oxidation³, low-temperature water-gas-shift (WGS) reaction⁴, methanol synthesis⁵ and so on. However, supported

copper-based catalyst is usually regarded as a low-temperature catalyst because copper species tend to agglomerate in high-temperature reactions and lead to the catalyst deactivation^{6,7}. How to anchor abundant and stable active copper sites on the support is the key to get excellent catalytic performance. In previous works, the sintering of catalysts always meant the loss of structural homogeneity and the severe deactivation⁸. **Instead, we used a simple deposition-precipitation preparation method and sintering strategy to construct supported copper cluster catalyst with very high loading (15 wt%) which was very stable and active at high temperatures and with reducing atmospheres. The strategy of constructing stable catalyst by partially sintering provides a reference for the preparation of efficient supported catalysts.**

For copper-based catalysts, due to the small difference in contrast in the HRTEM image between ceria and copper, it is difficult to accurately identify the structure of copper species in copper–ceria catalysts^{4,9–10}. Recently, Shen et al. explored the interfacial structures of copper–ceria catalysts **with a low copper loading (~3 wt.%)** after the treatments at relative low temperature under H₂ gas flow **for only 2 hours** (W. J. Shen et al, *Nat. Catal.* **2019**, 2, 334–341)¹². As shown in Figure R2, under mild treatment conditions, the structure of CeO₂ had no change, and the copper species could be stabilized on the support in the form of dispersed clusters. Therefore, it was expected that copper clusters would remain stable on the ceria under mild treatment conditions. However, in our work, we explored the structural transformation of copper–ceria catalyst after pretreatment and long-term reaction. **As shown in Figure R3, after pretreatment by H₂, the ceria nanorods were not sintered and the copper species with high copper loading (~15 wt.%) were dispersed as clusters on the support. When the catalyst underwent long-term RWGS reaction (for 70 and 240 h) under harsh conditions (600 °C, GHSV=400,000 mL·g_{cat}⁻¹·h⁻¹), the highly dispersed copper clusters was still anchored on the partially sintered ceria nanorods, which has never been reported in previous research work.** This finding clearly reveals the structure of partially sintered copper–ceria catalyst and illustrates

the copper–ceria catalyst can be used not only in low-temperature reactions but also have application potential in high-temperature hydrogenation reactions.

Figure R2. (a) A HAADF-STEM image of copper clusters dispersed on a ceria rod. (b, c) Atom-resolved HAADF-STEM images of copper clusters on ceria rods; the inset in b is an enlarged image of a selected bilayer that shows the arrangements of copper atoms within the copper cluster and at the copper–ceria interface. (from W. J. Shen et al, *Nat. Catal.* **2019**, 2, 334–341.)

Figure R3. (a) HAADF-STEM images of the 15CuCe catalyst after different treatments. (a–b) H₂ pretreatment. (c–d) 70 h RWGS reaction. (e–f) 240 h RWGS reaction.

2. How to construct catalyst with excellent activity and high stability to catalyze the RWGS reaction under very harsh reaction conditions?

The RWGS reaction is endothermic, a high working temperature is usually required to facilitate the equilibrium conversion of CO₂, however high temperature can easily cause the severe deactivation of catalysts. **Recently, although much works on the RWGS catalysts have been reported, the catalyst which can combine high activity and stability at high temperature has not been developed.** For example, Liu et al. reported the *in situ* visualization of the dynamic surface alloying in Ni@Au core-shell NPs under CO₂ hydrogenation conditions (W. Liu et al, *Nat. Catal.* 2020, 3, 411–417)¹³. But as shown in Figure R4, under the operating temperature of 600 °C and at a space velocity of **only ~60,000 ml g_{cat}⁻¹ h⁻¹**, the CO₂ conversion of this catalyst was **only 18%**. **In our work, the optimal 15CuCe catalyst showed the activity of 146.6 mol_{CO2}·g_{cat}⁻¹·s⁻¹ at 600 °C, which far exceeded that of other catalysts reported in literatures (Figure 4 and Table R1). Meanwhile, our copper–ceria catalyst also exhibited solid stability in the long-term test (for 240 h), which indicated that the partially sintered catalyst has indeed achieved a combination of excellent activity and high stability.**

Figure R4. The catalytic performance of the Ni@Au/SiO₂ catalyst for CO₂ hydrogenation at atmospheric pressure (24 vol% CO₂ + 72 vol% H₂ + 4 vol% Ar **at a space velocity of ~60,000 ml g_{cat}⁻¹ h⁻¹**). (from W. Liu et al, *Nat. Catal.* 2020, 3, 411–417.)

Figure R5. Comparison of the CO₂ conversion rates over different catalysts (see Table R1 for more details).

Table R1. Comparison of CO₂ Conversion Rates and CO Selectivity for the as-prepared and Literature Reported catalysts.

Catalyst	H ₂ :CO ₂	Temperature ^a	Pressure ^b	Rate ^c	CO selectivity ^d	Ref
15CuCe	3:1	600	0.1	146.6	100	this work
15CuAl	3:1	600	0.1	3.0	100	this work
Cu/CeO ₂ -hs	3:1	600	0.1	42.5	100	14
NiAu/SiO ₂	3:1	600	0.1	3.2	95	13
4Cu-Al ₂ O ₃	2:1	600	0.1	17.9	100	15
Cu/ β -Mo ₂ C	2:1	600	0.1	47.7	99.2	16
Cu-Fe/SiO ₂	1:1	600	0.1	11.9	100	6
NiCe/Zr	3:1	550	0.1	33.3	99.5	17
15CuCe	3:1	500	0.1	52.2	100	this work
In ₂ O ₃ -CeO ₂	1:1	500	0.1	2.98	100	18
K ₈₀ -Pt/L	1:1	500	0.1	2.22	100	19
Ni-in-Cu	3:1	500	0.1	3.95	100	20
CuSiO/CuO _x	3:1	500	0.1	3.18	100	21
TiO ₂ /Cu	3:1	500	0.1	1.78	N/A	21
SiO ₂ /Cu	3:1	500	0.1	1.11	N/A	21

Cu–Zn–Al	2:1	500	0.1	26.1	100	16
Cu/ β -Mo ₂ C	2:1	500	0.1	37.9	99.0	16
Pt/TiO ₂	1:1	400	0.1	5.0	100	22
Pt/Al ₂ O ₃	1.4:1	400	N/A	0.16	N/A	23
Ni/Mg(Al)O	3:1	450	0.1	0.5	66.7	24
Fe–CeO ₂	4:1	400	0.1	0.65	100	25

Notes: (a) °C, (b) MPa, (c) 10⁻⁵molCO₂/g_{cat}/s, (d) %.

3. How does oxygen vacancy in the catalyst play a role in the reaction process?

Defects in solid catalysts have an important effect on their catalytic activity. Although many previous reports have pointed out that oxygen vacancy was related to the catalytic activity of CO₂ reduction, **it is still unclear how the oxygen vacancy is involved in the reaction.** For example, A. Urakawa et al. proposed several possible pathways for the involvement of oxygen vacancies in the RWGS reaction (Figure R6), but there was not sufficient evidence to support the existence of these pathways (A. Urakawa et al, *ACS Catal.* **2018**, *8*, 7455–7457)²⁶. **In our work, the *in situ* Raman results clearly showed the relationship between oxygen vacancies and different reactant molecules. The surface oxygen vacancies were confirmed to be involved in the adsorption and the activation of CO₂ rather than the intrinsic bulk defects (Figure R7a–c). In addition, The DFT calculation further indicated that the synergistic effect of the active copper cluster and the adjacent oxygen vacancy was the key to promote the reaction process, while oxygen vacancy alone had limited ability to activate reactant molecules (Figure R7d–e).** Meanwhile, the combination of experimental results and DFT calculation demonstrated that the surface formate and carboxylic species might be the important reactive intermediates. The specific synergistic catalytic effect between copper cluster and oxygen vacancy also reflects that catalytic reactions are more likely to occur at the metal cluster-oxygen vacancy interfaces, which provides guidance for the design and development of efficient supported catalysts.

Figure R6. Proposed Mechanism of the RWGS Reaction for the Au/TiO₂ Catalyst. (from A. Urakawa et al, *ACS Catal.* **2018**, *8*, 7455–7457).

Figure R7. (a) *In situ* Raman under the RWGS reaction conditions for the 15CuCe catalyst. (b, c) *In situ* Raman of the 15CuCe catalyst with H₂/CO₂ switching under 300 °C and 500 °C, respectively. (d) Chemisorption of CO₂ on the 10Cu/CeO₂{111} surface. Five oxygen vacancies, named after V_O-A to V_O-E, were made comparisons, and the selected CO₂ was located close to V_O-A. (e) The adsorption energy that CO₂ was bound to V_O on the CeO₂{111} surface was obviously weaker than those of Cu dropped ceria sites.

In summary, the highlights of this work can be divided into the following points:

- (1) In a situation where sintered catalysts are often not favoured to catalysis, while we used **a simple deposition-precipitation method and *in situ* sintering strategy** to construct partially sintered copper–ceria catalyst with large number of stable copper cluster sites.
- (2) For the RWGS reaction, the partially sintered copper–ceria catalyst exhibited a **coexistence of excellent activity and high stability**, which raised the RWGS reaction activity to a new level.
- (3) The combination of experimental results and theoretical calculations revealed the **precise pathway for the involvement of oxygen vacancies in CO₂ activation** and highlighted the importance of the synergistic effect between active metal clusters and oxygen vacancies in the catalytic reaction.

We believe that our work is comprehensive, innovative and will be definitely helpful for researchers in not only catalysis, but also in general chemistry, materials sciences and energy-related disciplines. **The corresponding description has been added in line 27–29 on page 3, line 12–19 and 25–27 on page 4 in the revised manuscript (highlighted in yellow).** Thanks for the review's comment again.

To Reviewer #3

Comment: The authors have carefully revised the manuscript according to the reviewer's comments. They have seized all suggestions and significantly improved the manuscript.

Although the study is very comprehensive and explains the behavior and the structure of the catalyst with great detail, the studied catalyst suffers from somewhat limited technical relevance, since ceria nanorods are unlikely to be applied as support material in industrial processes. Nevertheless, the results are of sufficient novelty and detail to justify publication in Nature Communications.

I have the following comments that should be considered by the authors:

In Figure R1, it is unclear to which axis the shown bars refer.

In Figure R3 and R6, the differentiation between 2D and 3D clusters is not convincing. Due to the cylindrical shape of the nanorods, 3D clusters distributed on the surface of the nanorods could appear as flat or twodimensional (2D) depending on the viewing angle. Please provide more convincing results and arguments for the claimed 2D clusters.

Response: According to the reviewer's previous comments and suggestions, we added relative experiments and carefully revised the manuscript, which undoubtedly improved our work a lot. And thanks for the suggestions on our work again, we will reply to your comments one by one and look forward to your next comment.

Comment 1: In Figure R1, it is unclear to which axis the shown bars refer.

Response: In order to more clearly demonstrate the catalytic performance of the catalyst under different reaction gas compositions, we divided the original graph into two figures to show the CO₂ conversion and CO selectivity respectively. **The Figure R8 has been added as Figure S2 in the revised supporting information on page S7.** Thanks for the reviewer's valuable comments again.

Figure R8. Catalytic performances for the 15CuCe catalyst with H₂:CO₂ ratio of 2:1, 3:1, 4:1. (a) CO₂ conversion and (b) CO selectivity.

Comment 2: In Figure R3 and R6, the differentiation between 2D and 3D clusters is not convincing. Due to the cylindrical shape of the nanorods, 3D clusters distributed on the surface of the nanorods could appear as flat or twodimensional (2D) depending on the viewing angle. Please provide more convincing results and arguments for the claimed 2D clusters.

Response: The definition of 2D and 3D clusters is based on the observed shapes of the clusters on CeO₂ nanorods. As shown in Figure R9 and Figure R10, the diameter of 3D clusters is around 2 nm, while the thickness and width of 2D clusters are less than 1 and 3.5 nm. As the reviewer mentioned, we indeed observed 3D clusters which appear as round shape on the surface of CeO₂ nanorods. Because atomic number of Cu is much smaller than that of Ce, the extra brightness introduced by 3D clusters on the surface of CeO₂ is relatively small but still visible, as indicated by red dashed circles. On the other hand, due to thin layer of 2D cluster (less than 1 nm), the introduced brightness on the surface of CeO₂ is almost invisible. If the incident electron beam is parallel to the plane of 2D clusters, the 2D cluster would appear as nanorods as indicated by two parallel dashed lines near the edge of CeO₂. The Figure R9 and R10 have been added as Figure S9 and Figure 3, respectively.

And the corresponding description has been shown in page 8, line 20 in the revised manuscript (highlighted in yellow).

Figure R9. (a, b, c, d) HAADF-STEM images of the pre-reduced 15CuCe catalyst.

Figure R10. (a) TEM image of the used 15CuCe catalyst after 240 h stability test. (b–d) STEM images and element mapping results of the used 15CuCe catalyst after 240 h stability. (e–h)

HAADF-STEM images of the used 15CuCe catalyst after 240 h RWGS reaction. (i) Enlarged image of the highlighted region in (h).

References

1. N. Nityashree, C.A.H. Price, L. Pastor-Perez, G.V. Manohara, S. Garcia, M.M. Maroto-Valer, T.R. Reina, *Appl. Catal. B* 261 (2020) 118241.
2. Q. Zhang, L. Pastor-Pérez, W. Jin, S. Gu, T.R. Reina, *Appl. Catal. B* 244 (2019) 889–898.
3. A. Davó-Quiñero, M. Navlani-García, D. Lozano-Castelló, A. Bueno-López, J. A. Anderson, *ACS Catal.* 6 (2016) 1723–1731.
4. W.Z. Yu, W.W. Wang, C. Ma, S.Q. Li, K. Wu, J.Z. Zhu, H.R. Zhao, C.H. Yan, C.J. Jia, *J. Catal.* 402 (2021) 83–93.
5. S. Kattel, P.J. Ramírez, J.G. Chen, J.A. Rodriguez, P. Liu, *Science* 355 (2017) 1296–1299.
6. C.S. Chen, W.H. Cheng, S.S. Lin, *Applied Catalysis A: General.* 257 (2004) 97–106.
7. C.S. Chen, J.H. Lin, J.H. You, C.R. Chen, *J. Am. Chem. Soc.* 128 (2006) 15950–15951.
8. C.T. Campbell, S.C. Parker, D.E. Starr, *Science* 298 (2002) 811–814.
9. Y. Xie, J.F. Wu, G.J. Jing, H. Zhang, S.H. Zeng, X.P. Tian, X.Y. Zou, J. Wen, H.Q. Su, C.J. Zhong, P.X. Cui, *Appl. Catal. B* 239 (2018) 665–676.
10. D. Vovchok, C. Zhang, S. Hwang, L.Y. Jiao, F. Zhang, Z.Y. Liu, S.D. Senanayake, J.A. Rodriguez, *ACS Catal.* 10 (2020) 10216–10228.
11. S.C. Yang, S.H. Pang, T.P. Sulmonetti, W.N. Su, J.F. Lee, B.J. Hwang, C.W. Jones, *ACS Catal.* 8 (2018) 12056–12066.
12. A.L. Chen, X.J. Yu, Y. Zhou, S. Miao, Y. Li, S. Kuld, J. Sehested, J.Y. Liu, T. Aoki, S. Hong, M.F. Camellone, S. Fabris, J. Ning, C.C. Jin, C.W. Yang, A. Nefedov, C. Wölfl, Y.M. Wang, W.J. Wen, *Nature. Catal.* 2 (2019) 334–341.
13. X.B. Zhang, S.B. Han, B.E. Zhu, G.H. Zhang, X.Y. Li, Y. Gao, Z.X. Wu, B. Yang, Y.F. Liu, W. Baaziz, O. Ersen, M. Gu, J.T. Miller, W. Liu, *Nature. Catal.* 3 (2020) 411–417.
14. Y.D. Zhang, L. Liang, Z.Y. Chen, J.J. Wen, W. Zhong, S.B. Zou, M.L. Fu, L.M. Chen, D.Q. Ye, *Appl. Surf. Sci.* 516 (2020) 146035.

15. A.M. Bahmanpour, F. Héroguel, M. Kılıç, C.J. Baranowski, L. Artiglia, U. Röthlisberger, J.S. Luterbacher, O. Kröcher, *ACS Catal.* 9 (2019) 6243–6251.
16. X. Zhang, X.B. Zhu, L.L. Lin, S.Y. Yao, M.T. Zhang, X. Liu, X.P. Wang, Y.W. Li, C. Shi, D. Ma, *ACS Catal.* 7 (2017) 912–918.
17. P.C. Zonetti, S. Letichevsky, A.B. Gaspar, E.F. Sousa-Aguiar, L.G. Appel, *Applied Catalysis A: General.* 475 (2014) 48–54.
18. W. Wang, Y. Zhang, Z.Y. Wang, J.M. Yan, Q. F. Ge, C.J. Liu, *Catal. Today.* 259 (2016) 402–408.
19. X.L. Yang, X. Su, X.D. Chen, H.M. Duan, B. L. Liang, Q.G. Liu, X.Y. Liu, Y.J. Ren, Y.Q. Huang, T. Zhang, *Appl. Catal. B* 216 (2017) 95–105.
20. L.X. Wang, E. Guan, Z.Q. Wang, L. Wang, Z.M. Gong, Y. Cui, Z.Y. Yang, C. T. Wang, J. Zhang, X.J. Meng, P.J. Hu, X.Q. Gong, B.C. Gates, F.S. Xiao. *ACS Catal.* 10 (2020) 9261–9270.
21. Y. Yu, R.X. Jin, J. Easa, W. Lu, M. Yang, X.C. Liu, Y. Xing, Z. Shi, *Chem. Commun.* 55 (2019) 4178–4181.
22. X.D. Chen, X. Su, H.M. Duan, B.L. Liang, Y.Q. Huang, T. Zhang, *Catal. Today.* 281 (2017) 312–318.
23. S.S. Kim, H.Y. Lee, S.C. Hong, *Applied Catalysis A: General.* 423–424 (2012) 100–107.
24. M.T. Rodrigues, P.C. Zonetti, O.C. Alves, E.F. Sousa-Aguiar, L.E.P. Borges, L.G. Appel, *Applied Catalysis A: General.* 543 (2017) 98–103.
25. B.C. Dai, G.L. Zhou, S.B. Ge, H.M. Xie, Z.J. Liao, G.Z. Zhang, K. Xiong, *The Canadian Journal of Chemical Engineering.* 95 (2017) 634–642.
26. L.F. Bobadilla, J.L. Santos, S. Ivanova, J.A. Odriozola, A. Urakawa, *ACS Catal.* 8 (2018) 7455–7467.

REVIEWERS' COMMENTS

Reviewer #1 (Remarks to the Author):

The authors have carefully revised the manuscript according to the reviewer's comments.

The results are of sufficient and detail to justify publication in Nature Communications.

Reviewer #3 (Remarks to the Author):

The authors have compiled a comprehensive answer on the critical comment about the claimed novelty of their contribution. They could show in which aspects their new catalyst and study of its structure is novel and a significant advancement. Moreover, they have addressed all detailed comments convincingly.

Responses to the Reviewers' Comments and the Corresponding Revisions

Reviewer #1:

Comment: The authors have carefully revised the manuscript according to the reviewer's comments. The results are of sufficient and detail to justify publication in Nature Communications.

Response: Thanks for reviewer's comments. According to reviewer's comments, we have carefully revised and supplemented the manuscript, which undoubtedly greatly improved the quality of our research work. Thanks for the reviewer again.

Reviewer #3:

Comment: The authors have compiled a comprehensive answer on the critical comment about the claimed novelty of their contribution. They could show in which aspects their new catalyst and study of its structure is novel and a significant advancement. Moreover, they have addressed all detailed comments convincingly.

Response: The reviewer's professional and detailed comments and suggestions are of great help to the improvement of our research work. According to the reviewer's comments, the quality of our manuscript has been greatly improved compared to the original version. Thanks for the reviewer again.